# Halftone-encoded 4D printing of stimulus-reconfigurable binary domains for cephalopod-inspired synthetic smart skins

Haoqing Yang[1], Haotian Li[1], Juchen Zhang[1], Tengxiao Liu[2], H. Jerry Qi [3] & Hongtao Sun [1,2,4]

Cephalopods exhibit versatile control over their optical appearance, texture, and shape for adaptive camouflage and signaling. Achieving such multi-feature dynamic control in synthetic materials remains a significant challenge. Here, we introduce a halftone-encoded 4D printing method that enables simultaneous and programmable control over optical appearance, mechanical properties, surface texture, and shape transformation within a single smart hydrogel film in response to various external stimuli (*e.g.*, temperature, solvents, and mechanical stress)—a capability beyond existing synthetic materials. By encoding halftone binary patterns composed of highly crosslinked ("1") and lightly crosslinked ("0") domains, we spatially regulate localized polymer-solvent interactions and microstructural heterogeneities. The interplay, arrangement, and integration of these binary domains collectively dictate macroscale multifunctionality within a single material system. This binary encoding approach offers a simple yet powerful platform for designing multifunctional synthetic materials with complex, reconfigurable behaviors, unlocking opportunities in soft robotics, adaptive surface engineering, and secure information storage.

Soft-bodied cephalopods, such as squids, cuttlefish, and octopuses, possess a remarkable dynamic change in multiple features, including skin color, optical transmission, texture, and body shape (i in Fig. 1a)[1–4]. This adaptability allows them to blend seamlessly into their environment, evade predators, and communicate with each other. For example, the skin coloration of these organisms is primarily controlled by specialized neuromuscular organs called chromatophores, which contain pigment sacs encircled by radial muscles. When stimulated by neural signals, the radial muscles stretch the chromatophores, expanding the pigment sacs from small, punctate chromatocytes into wide, flat discs of color (ii in Fig. 1a)[4,5]. This rapid transformation allows for swift color changes in response to environmental cues, aiding camouflage and communication. Additionally, cephalopods use muscular hydrostats to dynamically adjust their

skin texture, shifting from smooth to highly textured surfaces by forming papillae (iii in Fig. 1a). These texture changes further enhance their ability to visually blend into surroundings[6,7]. They also employ circular and longitudinal muscles to control overall body shape[2]. This intricate system of nerves and muscles grants soft-bodied organisms the remarkable ability to simultaneously alter their optical appearance, surface texture, and shape.

However, achieving such refined, multi-faceted control in synthetic materials remains a significant challenge. Nanocomposites offer tunable optical and mechanical properties but typically lack dynamic reconfigurability[8–10]. Shape memory polymers (SMPs) enable programmable shape changes yet provide limited optical functionality[11–15]. Liquid crystal elastomers (LCEs) combine optical anisotropy with stimuli-responsiveness but struggle to achieve multi-modal control[16,17].

[1]The Harold & Inge Marcus Department of Industrial & Manufacturing Engineering, The Pennsylvania State University, University Park, PA, USA. [2]Department of Biomedical Engineering, The Pennsylvania State University, University Park, PA, USA. [3]The George W. Woodruff School of Mechanical Engineering, Georgia Institute of Technology, Atlanta, GA, USA. [4]Materials Research Institute (MRI), The Pennsylvania State University, University Park, PA, USA. ✉e-mail: qih@me.gatech.edu; hongtao.sun@psu.edu

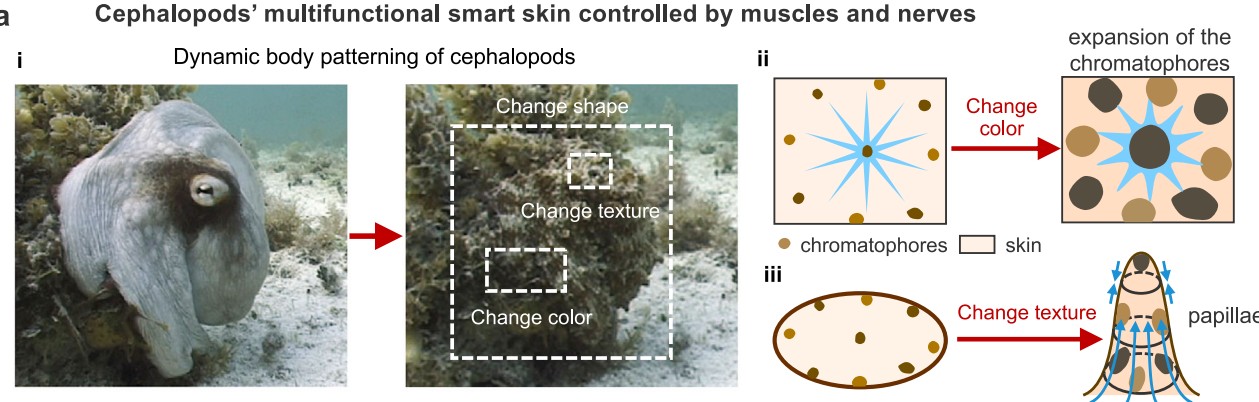

**a** Cephalopods' multifunctional smart skin controlled by muscles and nerves

**b** 4D printing cephalopod-inspired smart hydrogels with programmable halftone patterns for multi-functions

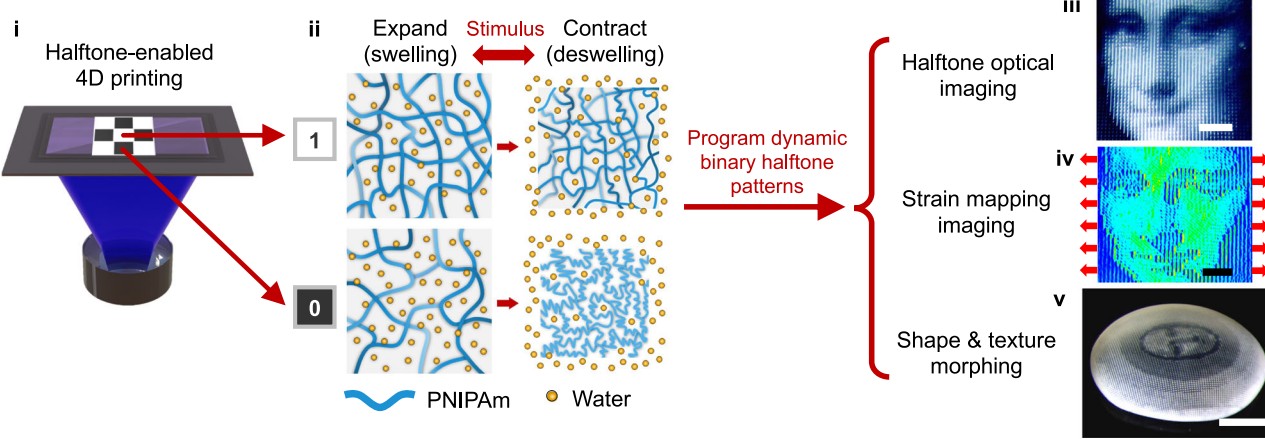

**Fig. 1 | Halftone-enabled 4D printing of multifunctional smart hydrogel "skins". a** Multifunctional smart skin in cephalopods, controlled by a complex neuromuscular system. Octopus vulgaris uses dynamic body patterning to modulate skin color, texture, and body shape for adaptive camouflage (i). Two photographs in (i) reproduced with permission from ref. 2, Elsevier. The schematic illustrates mechanisms by which chromatophores and papillae are controlled to achieve color and texture changes (ii and iii). **b** 4D-printed smart hydrogel "skins" featuring programmable binary halftone domains to regulate multiple dynamic properties, including optical transmittance, mechanical properties, and 2D-to-3D shape transformation during the swelling (expansion)-deswelling (contraction) cycle. White pixels ("1") correspond to highly crosslinked domains (120-s UV exposure), while black pixels ("0") represent lightly crosslinked domains (50-s UV exposure). Scale bars: 2 mm for optical (iii) and strain mapping images (iv); 5 mm for the shape-morphed non-Euclidean structure (v) in (**b**).

Smart hydrogels have emerged as promising alternatives due to their intrinsic stimulus-responsiveness and tunable optical properties[18–23]. Nonetheless, these synthetic material systems still fall short of realizing simultaneous and coordinated control over diverse dynamic features within a single construct (see Supplementary Table 1)[8–23].

Inspired by the dynamic behaviors of cephalopods, we have developed smart hydrogel "skins" capable of simultaneously tuning optical and mechanical properties, textures, and programmable shape transformations in response to various external stimuli, including temperature, solvents, and mechanical stress. This is achieved through a halftone-enabled 4D printing technique that encodes binary halftone patterns composed of highly crosslinked "1" domains and lightly crosslinked "0" domains into smart hydrogels (i and ii in Fig. 1b). Specifically, this approach leverages the precise arrangement of binary halftone patterns with varying domain sizes and spacing to simulate continuous tones (e.g., grayscale). By encoding these simple binary patterns, we manipulate localized stimulus-reconfigurable optical transmission, mechanical properties, and deformation within binary "1" or "0" domains, whose interaction, arrangement, and integration collectively dictate the overall dynamic complexity across hierarchical scales in a single material system.

For example, much like the neuromuscular-controlled patterning observed in cephalopods, our halftone-encoded smart hydrogels dynamically manipulate distinct transitions in optical transmittance within local binary domains by altering the surrounding solvents or temperatures as they transition through the lower critical solution temperature (LCST) during the swelling-deswelling cycle. The precise arrangement and integration of these local dynamic optical domains enable the overall hydrogel "skins" to conceal or reveal high-resolution, high-contrast halftone images in response to solvent and temperature changes (iii in Fig. 1b). Beyond optical control, the strategic organization of these local binary domains also tailors heterogeneous mechanical responses under small external forces, which can be decrypted using full-field strain mapping via real-time digital image correlation (DIC) analysis (iv in Fig. 1b). Moreover, by simultaneously encoding optical transitions and deswelling-induced deformation into local binary domains, we integrate high-resolution image information with programmable in-plane growth functions within a single hydrogel "skin." Upon external stimulation, the encrypted image information is retrieved as the material morphs into prescribed non-Euclidean 3D configurations with controlled textures and Gaussian curvatures through out-of-plane deformation (v in Fig. 1b). Thus, this binary encoding mechanism provides a simple yet powerful platform for co-designing multifunctional synthetic materials with complex, reconfigurable behaviors—a capability beyond existing synthetic materials.

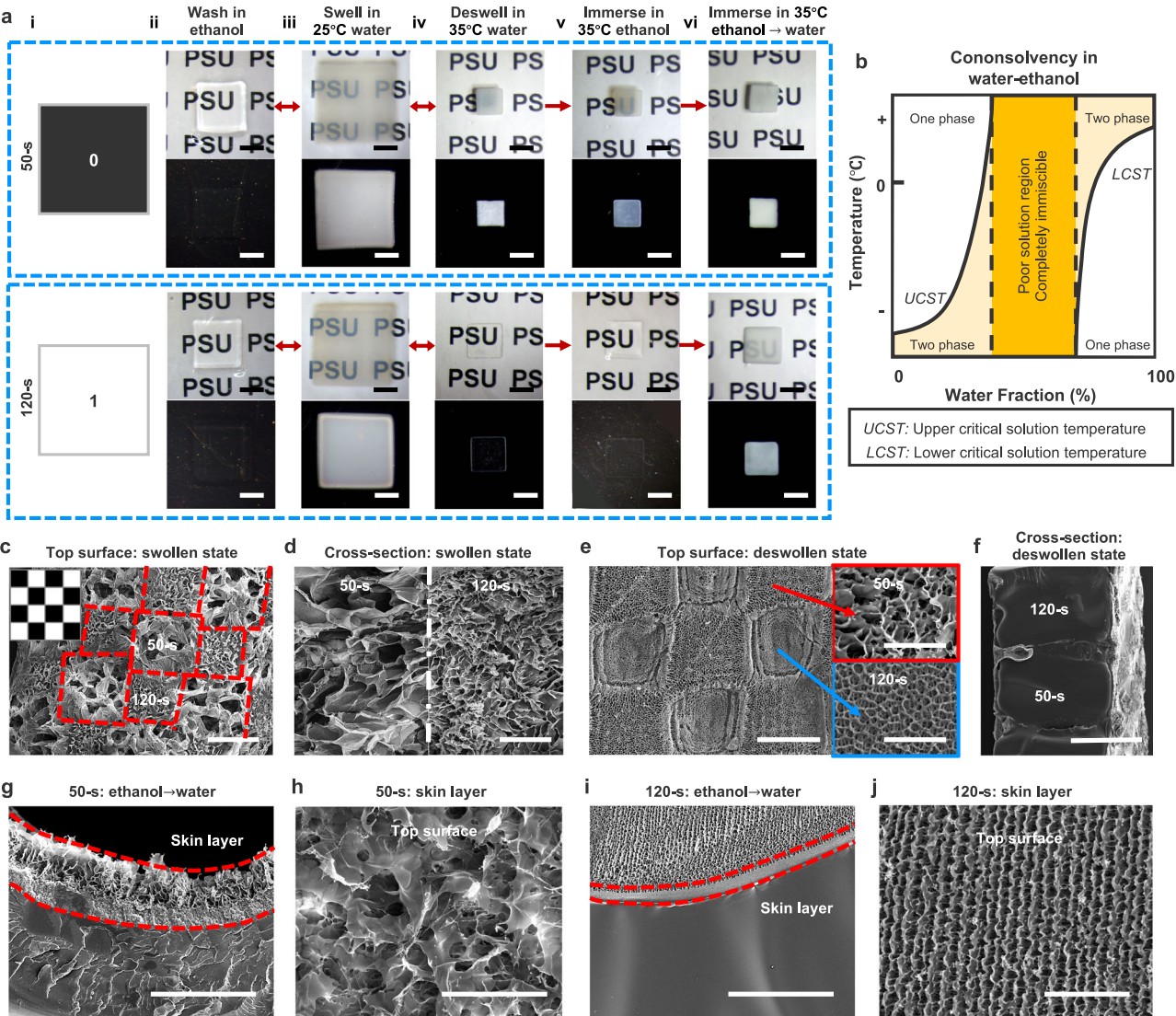

**Fig. 2 | Optical appearance and morphological characterization of hydrogel films under different swelling-deswelling states and solvent conditions. a** The optical appearance of homogeneous hydrogel films printed with low (50-s, "0") and high (120-s, "1") UV exposures, shown on white paper with text or against a black background. **b** Schematic phase diagram illustrating PNIPAm cononsolvency in a water-ethanol solvent system. Scanning electron microscopy (SEM) images showing the top view (**c**, **e**) and cross-sectional view (**d**, **f**) of hydrogel films with 50-s and 120-s checkered patterns, swollen in water at 25 °C (**c**, **d**) and deswollen in water at 35 °C (**e**, **f**). SEM images showing the cross-sectional view (**g**, **i**) and top view (**h**, **j**) of deswollen hydrogel films after immersion in ethanol for 15 s, followed by water immersion at 35 °C. Hydrogel films were freeze-dried prior to morphological characterization. Scale bars, 5 mm (**a**), 500 μm (**c**, **f**), 200 μm (**d**, **e**, **g**, **i**), 100 μm (**h**), 50 μm for selected domains in (**e**) and (**j**).

## Results

### 4D-printed stimulus-reconfigurable binary domains for spatially controlled optical appearance

To print 2D hydrogel films, we designed a custom fixed-height cell with an interior spacing of 450–550 μm, constructed on a Teflon FEP sub-strate within a digital light processing (DLP) 3D printer. The polymer precursor solution was injected into this setup and covered with a thin glass slide. In this study, we used photocurable poly(N-iso-propylacrylamide) (PNIPAm) hydrogels, crosslinked with a combina-tion of a long-chain crosslinker, poly(ethylene glycol) diacrylate (PEGDA), and a short-chain crosslinker, N, N'-Methylenebis(acryla-mide) (BIS), in an acetone-water solvent. Our study indicates that the PEGDA and BIS can simultaneously tune the crosslinking properties of the hydrogel film, but to different extents. Under lower curing time (50 seconds), the hydrogel network was primarily crosslinked by the long-chain PEGDA and presented a loose microstructure; while under high curing time (120 seconds), more short-chain BIS participated and

dominated the following crosslinking process, and enhanced the net-work density and strength[24–26]. By varying UV exposure times during printing, we controlled the degree of crosslinking in the hydrogels, resulting in distinct variations in optical transmittance, mechanical properties, and thermally induced growth (expansion and contraction) during the swelling-deswelling cycle. These differences stem from distinct dynamic transitions between intermolecular and intramole-cular interactions[27,28].

We first examined variations in the optical transmittance and morphology of hydrogel films printed with varying UV exposures under different swelling-deswelling states and solvent conditions (Fig. 2, Supplementary Figs. 1 and 2). The as-printed hydrogel films were washed with ethanol and ice water (0 °C) to remove unreacted monomers and photoinitiators. Interestingly, when immersed in ethanol, both highly and lightly crosslinked hydrogel films became transparent, revealing letters on white paper or appearing black against a black background (ii in Fig. 2a). This transparency arose from

the gradual diffusion of ethanol into the hydrogel, which disrupted water-polymer interactions due to the cononsolvency effect[29–32]. Cononsolvency effectively lowered the LCST (Fig. 2b), promoting network collapse and reducing light-scattering boundaries, thereby enhancing transparency.

When soaked in 25 °C water, both highly and lightly crosslinked hydrogels became translucent in their swollen state, exhibiting minimal differences in optical transmission (iii in Fig. 2a, and Supplementary Fig. 1a). This translucency was due to their porous structures (50-s and 120-s regions in Fig. 2c, d, and Supplementary Fig. 1b, c), where light scattering occurred at the interfaces between polymer-rich and water-rich phases. These interfaces disrupted transparency, making the hydrogels translucent in the swollen state.

However, upon heating to 35 °C (above the LCST), the films deswelled, exhibiting distinct optical appearances in their fully deswollen states: lightly crosslinked hydrogels (50-s) turned opaque; opacity gradually diminished with longer curing times (60-s – 110-s); and highly crosslinked hydrogels (120-s) became transparent (iv in Fig. 2a, and Supplementary Fig. 2). This progression generated a striking black-and-white visual contrast against a black background (iv in Fig. 2a, Supplementary Figs. 1d and 2). The distinct optical changes can be explained as follows. Shorter UV exposure (50 s) produced lightly crosslinked hydrogels with larger pores in the swollen state (50-s regions in Fig. 2c, d, and Supplementary Fig. 1b). This porous structure allowed greater microphase separation between water-rich and polymer-rich regions during deswelling[33–35]. As water-filled voids collapsed, the hydrogel formed a porous skin layer and a dense interior (50-s regions in Fig. 2e, f, and Supplementary Fig. 1e, f), leading to pronounced light scattering, an opaque appearance, and greater shrinkage (50-s hydrogel, iv in Fig. 2a)[10,33–38].

In contrast, longer UV exposure (120 s) yielded a densely crosslinked network with smaller pores in the swollen state (120-s regions in Fig. 2c, d, and Supplementary Fig. 1c) due to increased crosslinking by PEGDA and BIS. Upon deswelling, this network collapsed into a dense interior with a relatively smoother surface featuring ripple-like patterns (120-s regions in Fig. 2e, f, and Supplementary Fig. 1e, f). The reduced surface porosity minimized light scattering, enhancing transparency (120-s hydrogel, iv in Fig. 2a). When preparing freeze-dried hydrogel samples for SEM characterizations, ice crystal formation may also contribute to pore formation. However, the considerable differences in porous morphologies primarily result from UV-regulated photopolymerization and the associated phase separation during swelling-deswelling. These observations highlight the distinct swelling-deswelling behaviors of hydrogel films printed under different UV exposure times. Higher crosslinking density also improved mechanical properties but reduced the hydrogel's deformability during swelling and deswelling[28].

Upon immersion in ethanol at 35 °C for approximately 15 s, the lightly crosslinked deswollen hydrogel film became translucent again, while the highly crosslinked deswollen hydrogel film remained transparent (v in Fig. 2a). Subsequent immersion in 35 °C water rendered both films opaquer (vi in Fig. 2a). This transformation resulted from cononsolvency-induced phase separation (Fig. 2b) and the formation of thicker porous skin layers (Fig. 2g–j). Specifically, both hydrogel films developed a dense interior with a porous surface layer (Fig. 2g, i) following ethanol and water immersion, leading to pronounced light scattering and an opaque appearance. Despite their shared opacity, the surface morphologies differed between the lightly (Fig. 2g, h) and highly crosslinked films (Fig. 2i, j), with noticeably thicker skin layers compared to those in their fully deswollen state in 35 °C water (Fig. 2e, f, and Supplementary Fig. 1e, f).

Furthermore, we developed a digital halftone-encoded 4D printing technique to achieve precise spatiotemporal control over photopolymerization within smart hydrogel films. This method utilizes a dynamic mask generated by the Digital Micromirror Device (DMD) in a DLP 3D printer (resolution: ~50 μm/pixel) to encode halftone patterns. Specifically, binary halftone patterns spatially define lightly crosslinked regions (e.g., 50-s regions: black "0" pixels) and highly crosslinked regions (e.g., 120-s regions: white "1" pixels) within a single printed hydrogel film. As the film deswells in response to thermal stimuli, the binary regions display distinct optical appearances due to variations in local optical transmittance.

To achieve varying grayscale levels for continuous-tone imaging, we implemented two halftoning algorithms, frequency-modulated (FM) and amplitude-modulated (AM) halftoning methods[39], to generate 10 grayscale levels within a 6 × 6-pixel unit matrix (top panels, Fig. 3a, b). In the FM method, the highly crosslinked white domain "1" is fixed at a 2 × 2-pixel domain size, while the frequency of additional "1" domains increases from G0 to G9, gradually brightening the tone and increasing total UV exposure per unit matrix. In contrast, the AM method varies the size of the "1" domain while maintaining a constant frequency within the unit matrix across different grayscale levels.

Although these two methods generate different halftone patterns, grayscale levels can be equalized by balancing the total number of binary pixels within each unit matrix. In the halftone matrix, lower cumulative exposure, represented by darker tones (e.g., more black pixels at G0-G2 in the top panels of Fig. 3a, b), results in greater opacity in deswollen hydrogel films (bottom panels of Fig. 3a, b). Conversely, higher cumulative exposure, represented by brighter tones (e.g., more white pixels at G6-G9), yields more transparent regions that appear black against a black background. This results in an inverse white-to-black visual relationship between the designed halftone patterns and the optical appearance of the printed hydrogel film in the deswollen state. Notably, the printed areas of lightly crosslinked "0" domains appear smaller than their designed ones due to free-radical diffusion and overcuring in locally low-dose regions[40], as well as their greater shrinkage in the deswollen state compared to the highly crosslinked "1" domains (e.g., area dimensions in the deswollen state: $A_{50-s} < A_{120-s}$, iv in Fig. 2a)[28].

To demonstrate the dynamic graphic display enabled by halftone patterning, we used the Mona Lisa as an example. A 7000 × 7000-pixel RGB image of the Mona Lisa (Fig. 3c) was first converted into a 120 × 120-pixel grayscale image (Fig. 3d). Grayscale values spanning from 10 to 220 were grouped into ten bins (Supplementary Fig. 3), each represented by a corresponding 6 × 6-pixel halftone unit matrix, with grayscale levels increasing from G0 to G9. These halftone-encoded grayscale levels (e.g., G0-G9) were defined using either the FM or AM method (Fig. 3a, b). This process transformed the 120 × 120-pixel grayscale image into a 720 × 720-pixel halftone image, as illustrated using the FM method (Fig. 3e and Supplementary Fig. 4a) or the AM method (Fig. 3f and Supplementary Fig. 5a).

Following halftone-regulated photopolymerization, the printed hydrogel films were washed with ethanol and ice water at 0 °C. During this washing process, the encrypted graphic information remained invisible in ethanol due to the film's transparency, appearing black against a black background (dark red dot-labeled image, Fig. 3g). However, upon immersion in ice water, the image gradually emerged within 60-100 seconds, revealing white contrasts in the brighter-tone regions (e.g., forehead, cheeks, chest, and sky, shown in yellow dot-labeled images, see Fig. 3g, Supplementary Fig. 6a, and Supplementary Movie 1).

This rapid transition in optical appearance were primarily driven by cononsolvency-regulated solvent-polymer interactions, inducing distinct transparency-to-opacity shifts across the local halftone binary domains. Specifically, as water diffused into the hydrogel films, replacing ethanol within the polymer network, the cononsolvency effect triggered a gradual optical transition, beginning at the surface skin layers. More importantly, the halftone-encoded hydrogel film demonstrated repeatable and reversible image concealment and revelation within 60–100 s through an ethanol-water immersion cycle

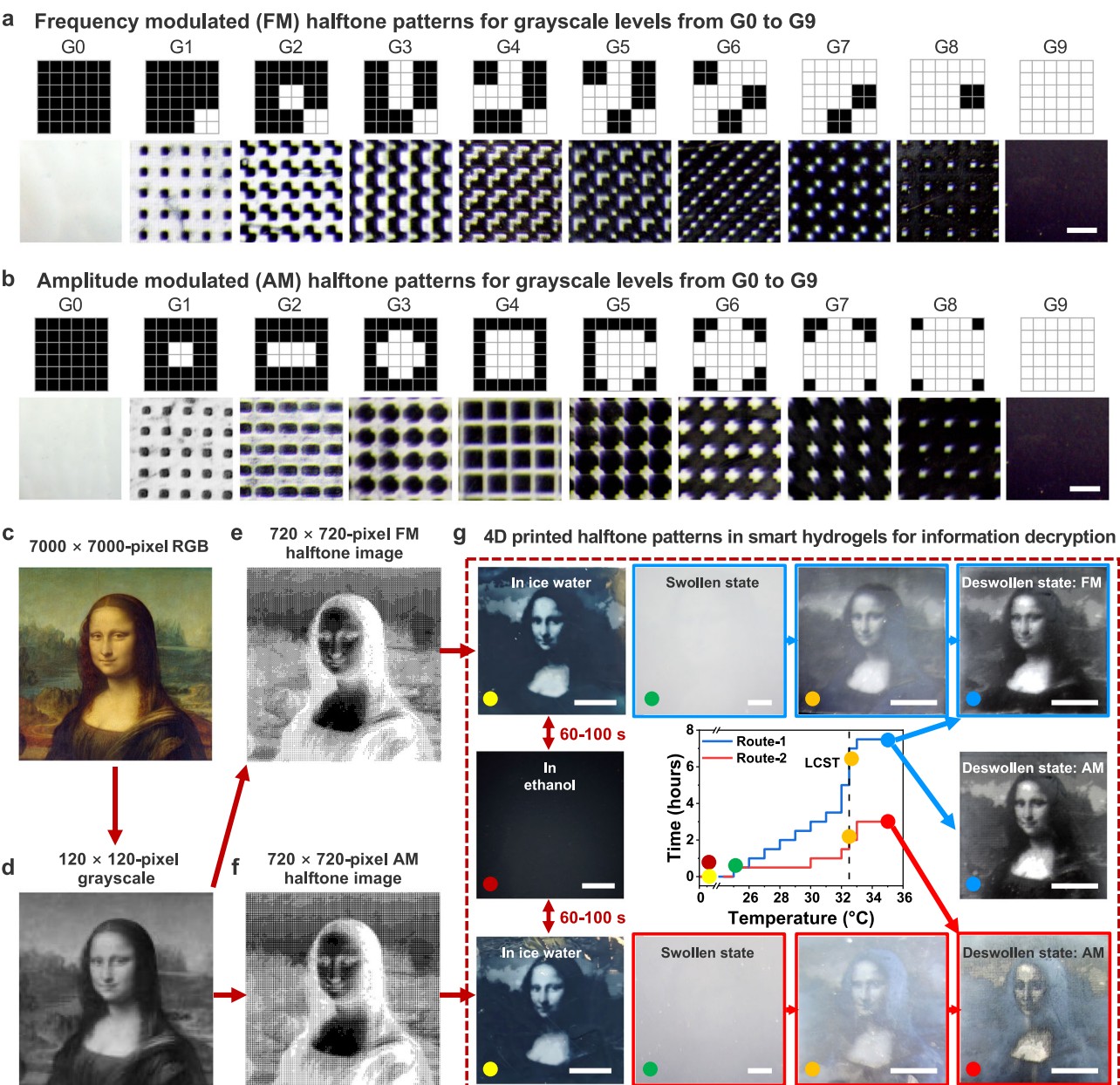

**Fig. 3 | Halftone-enabled 4D printing for dynamic information encryption-decryption.** Halftone patterns generated using the FM method (**a**) and AM method (**b**) within a 6 × 6-pixel unit matrix, displaying defined grayscale levels from G0 to G9 (top panels) and corresponding printed hydrogel films in the deswollen state (bottom panels). A 7000 × 7000-pixel RGB image of the Mona Lisa (**c**) converted to a 120 × 120-pixel grayscale image (**d**) and further encoded as 720 × 720-pixel halftone images using the FM method (**e**) or AM method (**f**). **c** "Mona Lisa" by Leonardo da Vinci, image courtesy of Rawpixel.com (https://www.rawpixel.com/image/7726890/image-mona-lisa-art-public-domain), licensed under the Creative Commons Attribution 4.0 International License (CC BY 4.0). **d** Adapted from the same source as (**c**), converted to grayscale image, licensed under the Creative Commons Attribution 4.0 International License (CC BY 4.0). **e** Adapted from the same source as (**c**), converted to halftone image, licensed under the Creative

Commons Attribution 4.0 International License (CC BY 4.0). **f** Adapted from the same source as (**c**), converted to halftone image, licensed under the Creative Commons Attribution 4.0 International License (CC BY 4.0). **g** 4D printed halftone-encoded hydrogel films for image concealment and revelation. The encrypted image information remained concealed in ethanol (dark red dot-labeled image) but gradually emerged within 60–100 s upon immersion in ice water, revealing contrast against a black background (yellow dot-labeled images) (Supplementary Movie 1); in the swollen state (25 °C water), the films appeared uniformly translucent against a black background (green dot-labeled images); image information was gradually revealed upon heating above the LCST through two protocols: during initial phase separation (orange dot-labeled images) and in the deswollen state (blue dot-labeled images for Route-1, red dot-labeled images for Route-2). Scale bars, 500 µm (**a**, **b**); 10 mm (**g**).

(Supplementary Fig. 6 and Supplementary Movie 1), underscoring its potential for dynamic information encryption-decryption.

These imaging contrasts disappeared once the films reached equilibrium in the swollen state at 25 °C in water, appearing uniformly translucent and light gray when placed against a black background (green dot-labeled images, Fig. 3g). This translucency, caused by light scattering, was attributed to the porous structures present in both

lightly crosslinked (50-s) and highly crosslinked (120-s) binary domains, as observed on the surface (Fig. 2c) and within the interior (Fig. 2d). Upon gradual heating to 35 °C in water, surpassing the LCST (blue Route-1, Fig. 3g), hydrogel films encoded using either the FM or AM method revealed a high-contrast halftone image of the Mona Lisa, accompanied by deswelling-induced shrinkage (blue dot-labeled images in Fig. 3g, Supplementary Figs. 4b, 5b, and Supplementary

Movie 2). These dynamic optical variations were driven by deswelling-induced phase separation, which led to differences in surface morphology and light scattering, particularly in the skin layers between highly and lightly crosslinked regions (Fig. 2e). To achieve faster dynamic decryption in water, an accelerated heating protocol was implemented across the LCST, reducing the decrypted time from 7.5 hours (blue Route-1, Figs. 3g) to 3 hours (red Route-2, Fig. 3g). Initially, the image appeared blurred due to rapid phase separation and retained water within the network. Maintaining the temperature above the LCST for an additional hour enhanced image clarity as most of the water gradually diffused out. Additionally, quenching the deswollen hydrogel film in ethanol rapidly erased the image, rendering the entire film transparent (Supplementary Fig. 7).

The FM halftoning method, with fine, randomly dispersed dots, excels in preserving detail and minimizing pattern artifacts, making it ideal for applications requiring high resolution and minimal visual interference (Supplementary Fig. 4). In contrast, AM halftoning produces smooth tonal gradients that are well-suited for contrast-rich images but may introduce Moiré patterns in high-detail graphics (Supplementary Fig. 5).

Thus, stimulus-reconfigurable halftone-encoded hydrogel films enable the concealment and revelation of high-contrast images for dynamic information encryption-decryption. The spatiotemporal control over optical appearance shifts in our hydrogel "skin", driven by the cononsolvency effect or thermally induced swelling-deswelling processes, emulates the dynamic patterning capabilities of cephalopods for communication and camouflage.

## Halftone pattern-regulated mechanical strain mapping for an additional layer of information encoding

Our hydrogel films exhibit not only customizable optical properties but also tunable mechanical characteristics that vary with their crosslinking networks (Supplementary Fig. 8a). Homogeneous hydrogel films exposed to UV light for 50 seconds (lightly crosslinked) and 120 seconds (highly crosslinked) displayed noticeable differences in mechanical properties in their deswollen states. The 50-s hydrogel film had a Young's modulus of 40.23 kPa and a strain at a break of 1102%, whereas the 120-s film exhibited a Young's modulus of 123.24 kPa and a strain at break of 206% (Supplementary Fig. 8a). The increase in Young's modulus with longer UV exposure is attributed to a higher degree of photopolymerization, which enhances crosslinking and stiffens the polymer network[28,41]. Mechanical testing of deswollen hydrogel films (in 35 °C water) encoded with FM and AM halftone patterns at varying grayscale levels (G0 to G9) revealed a progressive increase in Young's modulus, driven by the increasing proportion of highly crosslinked domains (Supplementary Fig. 8b–d).

Furthermore, the spatial arrangement of halftone patterns influenced localized mechanical responses under external forces, which can be characterized using real-time, full-field strain mapping via DIC analysis[28,42]. To investigate halftone-regulated mechanical behavior, we designed halftone patterns with a constant grayscale level (G3) but varied their orientation relative to the x-axis load direction: D1 (0°), D2 (45°), and D3 (90°) (Fig. 4a–c). In D1, where halftone patterns aligned with the load direction, minimal x-strain differences were observed between highly crosslinked "1" domains and lightly crosslinked "0" domains at 30% strain (Fig. 4a). At a 45° orientation, strain differentiation across local binary domains became noticeable (Fig. 4b) and was more pronounced when the halftone patterns were arranged perpendicular to the load direction (Fig. 4c). These orientations also influenced global mechanical properties, such as Young's modulus and strain at break, as shown in stress-strain curves (Fig. 4d).

This halftone pattern-regulated strain mapping provides an additional information encryption and decryption mechanism. As a demonstration, we encoded the letters "PSU" into a hydrogel film by rotating the local halftone pattern orientation within the letters by 30°

relative to the x-axis (Fig. 4e). In the deswollen state, the letters were nearly invisible in the optical image due to their grayscale levels matching the background (Fig. 4f). However, upon 5% stretching along the x-axis, the letters became discernible in the x-strain mapping due to distinct anisotropic strain responses (Fig. 4g). In contrast, stretching along the y-axis reduced visibility in the y-strain mapping due to a more uniform strain distribution across different halftone patterns (Fig. 4h).

Beyond anisotropy-driven strain mapping information, incorporating stiff cellular domains encoded at higher grayscale levels into softer films with lower grayscale levels induces localized heterogeneous subdomains under mechanical stretch, enriching strain mapping information[28]. Building on our previous work[28], we embedded a stiffer re-entrant honeycomb meta-structure (FM G9 halftone patterns) within a softer hydrogel film (FM G1 halftone patterns) (Fig. 4i, j). When stretched to 20 % along the y-axis, localized strain concentrations emerged at the centers of each cellular unit, forming periodic patterns in the y-strain mapping (Fig. 4k). Meanwhile, the x-strain mapping revealed distinct strain localization patterns due to strain-induced heterogeneous subdomains within the soft film (e.g., red and blue regions, Fig. 4l), as confirmed by strain profiles along a linear pathway X (Fig. 4m). These results demonstrate that the precise arrangement of halftone binary domains with different mechanical features enables the encoding and visualization of mechanical strain mapping information, adding an additional mode of information encoding.

We further explored how halftone pattern-regulated mechanical properties influence the decoding of the Mona Lisa portrait through full-field strain mapping. Instead of the previous FM and AM methods, we redefined grayscale levels from G0 (16.7% "1" pixels per 6 × 6-pixel unit matrix) to G9 (83.3% "1" pixels per unit matrix), using halftone patterns arranged in horizontal, vertical, or diagonal orientations (Fig. 5a, e, i). The corresponding optical halftone images of the Mona Lisa were clearly visualized, capturing intricate facial and background details regardless of pattern orientation (Fig. 5b, f, j, and Supplementary Fig. 9a–c). These high-contrast optical halftone images, encoded within the hydrogel films, demonstrated long-term stability, retaining clarity even after three days (Supplementary Fig. 9d, e).

Adjusting the halftone pattern orientation modulated anisotropic mechanical responses, influencing how the Mona Lisa appeared in strain mapping images and accentuating specific features. Under y-axis stretching, horizontally aligned halftone patterns highlighted features such as the eyes, nose tip, and mouth (Fig. 5c, d), while vertically aligned patterns emphasized the nose bridge and facial contours under x-axis stretching (Fig. 5g, h). However, when halftone patterns were aligned parallel to the stretch direction (e.g., vertically aligned patterns under y-axis stretching), strain mapping images displayed less discernable features (Supplementary Fig. 10). Diagonally arranged patterns exhibited stretch direction-dependent features, combining effects from both horizontal and vertical alignments (Fig. 5k, l). These decrypted features in strain mapping images arose from local mechanical variations dictated by the halftone binary domain arrangement.

To enhance graphic contrast in full-field strain mapping, we selectively replaced horizontally oriented patterns in darker or shadowed areas (e.g., nose, eyes, mouth, facial shadows, and hairs) with vertically aligned patterns at the same grayscale levels (left side of Fig. 5m, and Supplementary Fig. 11). When the deswollen hydrogel film was stretched along the x-axis, this adjustment sharpened facial features and contours in the strain mapping image (Fig. 5n, Supplementary Movie 3), creating clear interfaces between regions with distinct anisotropic mechanical behaviors. After immersing the deswollen film in ethanol for 15 seconds, followed by water at 35 °C, the film became opaquer, erasing the decrypted graphic information (right side of Fig. 5m). This transformation was attributed to the cononsolvency

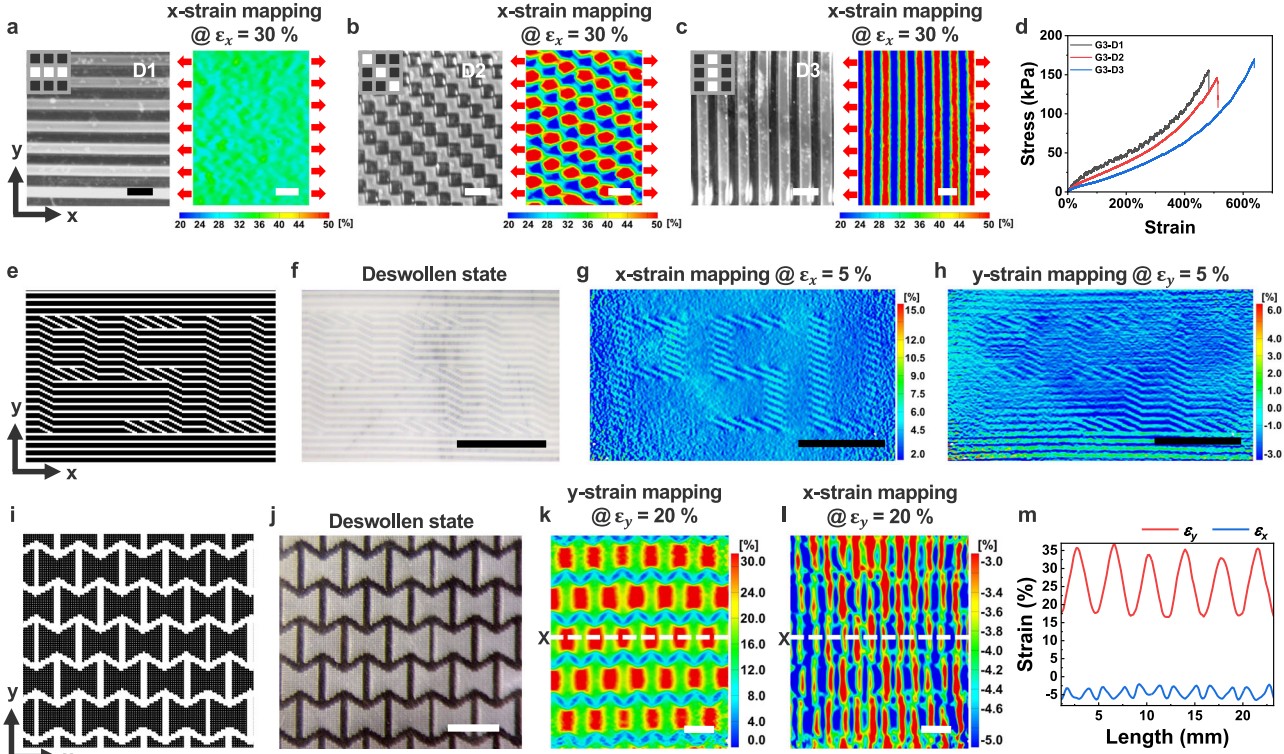

**Fig. 4 | Halftone pattern-regulated mechanical strain mapping via DIC analysis.** Optical images and x-strain mapping of deswollen hydrogel films (35 °C water) encoded with G3 (grayscale level) halftone patterns oriented at 0° (**a**), 45° (**b**), and 90° (**c**) under 30% strain $\varepsilon_x$. **d** Stress-strain curves of deswollen hydrogel films corresponding to (**a**–**c**). **e** Encryption of the letters "PSU" by rotating local halftone pattern 30° relative to the x-axis. **f** Optical image of the encrypted hydrogel film in its deswollen state. x-strain (**g**) and y-strain (**h**) mappings under 5% strain ($\varepsilon_x$ and $\varepsilon_y$) along the x- and y-axes, respectively. **i** Stiff re-entrant honeycomb meta-structure (G9 halftone) embedded in a soft film (G1 halftone). **j** Optical image of the meta-patterned film in its deswollen state. y-strain (**k**) and x-strain (**l**) mappings under y-axis loading. **m** Strain profiles along a marked linear pathway X. Scale bars, 500 μm (**a**–**c**), 5 mm (**f**–**h**, **j**–**l**).

effect and pore formation in the surface skin layers (Fig. 2g–j), inducing pronounced light scattering and opacity. Although the hydrogel film no longer displayed the encrypted halftone image, the strain mapping image preserved a high-contrast Mona Lisa representation, attributed to the distinct mechanical responses across the local binary domains (Fig. 5o, and Supplementary Movie 4).

Additionally, we demonstrated multi-layered information encryption and decryption within a single hydrogel film. For example, a second layer encoding of the letter "M," composed of diagonally aligned halftone patterns, was embedded within a background of horizontally oriented patterns. Due to its identical grayscale level with the forehead region, the letter remained nearly invisible in the deswollen film (Fig. 5p). However, in the strain mapping image, it became discernible due to distinct localized anisotropic mechanical responses in the letter region (Fig. 5q). The co-design of halftone pattern-regulated optical and mechanical features within a single hydrogel film provides a complementary approach for high-contrast, multi-modal, multi-layered information encryption and decryption, offering enhanced security compared to previously reported cryptology technologies (Supplementary Table 1)[8–23].

## Simultaneous control of optical appearance, shape transformation and surface texture

Beyond their stimulus-reconfigurable optical and mechanical properties, our halftone-encoded smart hydrogel films exhibit controlled in-plane growth, such as deswelling-induced shrinkage (Supplementary Movie 2), allowing simultaneous regulation of dynamic optical appearance and shape transformation-mimicking versatile dynamic behaviors observed in soft-bodied cephalopods. Specifically, lightly crosslinked hydrogels with larger pores absorb more water in the

swollen state, leading to greater shrinkage upon deswelling compared to highly crosslinked hydrogels with lower porosity (e.g., 50-s versus 120-s, from iii to iv in Fig. 2a)[28]. Inspired by the shape-morphing behavior of cephalopod muscles, we encoded halftone-regulated growth functions into 2D thermo-responsive hydrogel films, enabling precise control over nonuniform, in-plane deswelling-induced deformation. By modulating UV curing times across local binary domains, we generated halftone-regulated grayscale gradients that spatially control in-plane deformation gradients within a single layer-structure. Unlike conventional active-passive bilayer designs, which rely on bending mechanisms in shape-morphing composite materials[43], our approach enables the transformation of 2D single-layer films into predetermined 3D configurations through out-of-plane deformation during the swelling-deswelling cycle.

To derive the growth functions dictating 2D-to-3D shape transformations, we studied the relationships among halftone-encoded grayscale levels (G1-G9), their corresponding areal deswelling ratios ($A_{35°C}/A_O$), and the relative radius of concentric rings ($r/R$) (Fig. 6a, c). Using the FM method to encode grayscale levels from G1 to G9, we achieved areal deswelling ratios ($A_{35°C}/A_O$) ranging from approximately 0.3 to 0.5, as depicted by the projected blue curves (Fig. 6a, c). These halftone-encoded grayscale levels regulated deswelling-induced deformations in a manner similar to those observed under varying curing times (Supplementary Fig. 12). The spatial distribution of grayscale levels, governed by the variable $r/R$ (Fig. 6b, d, and Supplementary Fig. 13a, c), and their corresponding areal deswelling ratios ($A_{35°C}/A_O$) were programmed to define growth functions (projected red curves in Fig. 6a, c). These encoded functions precisely controlled the 2D-to-3D shape morphing process, enabling the formation of prescribed 3D configurations[24,28].

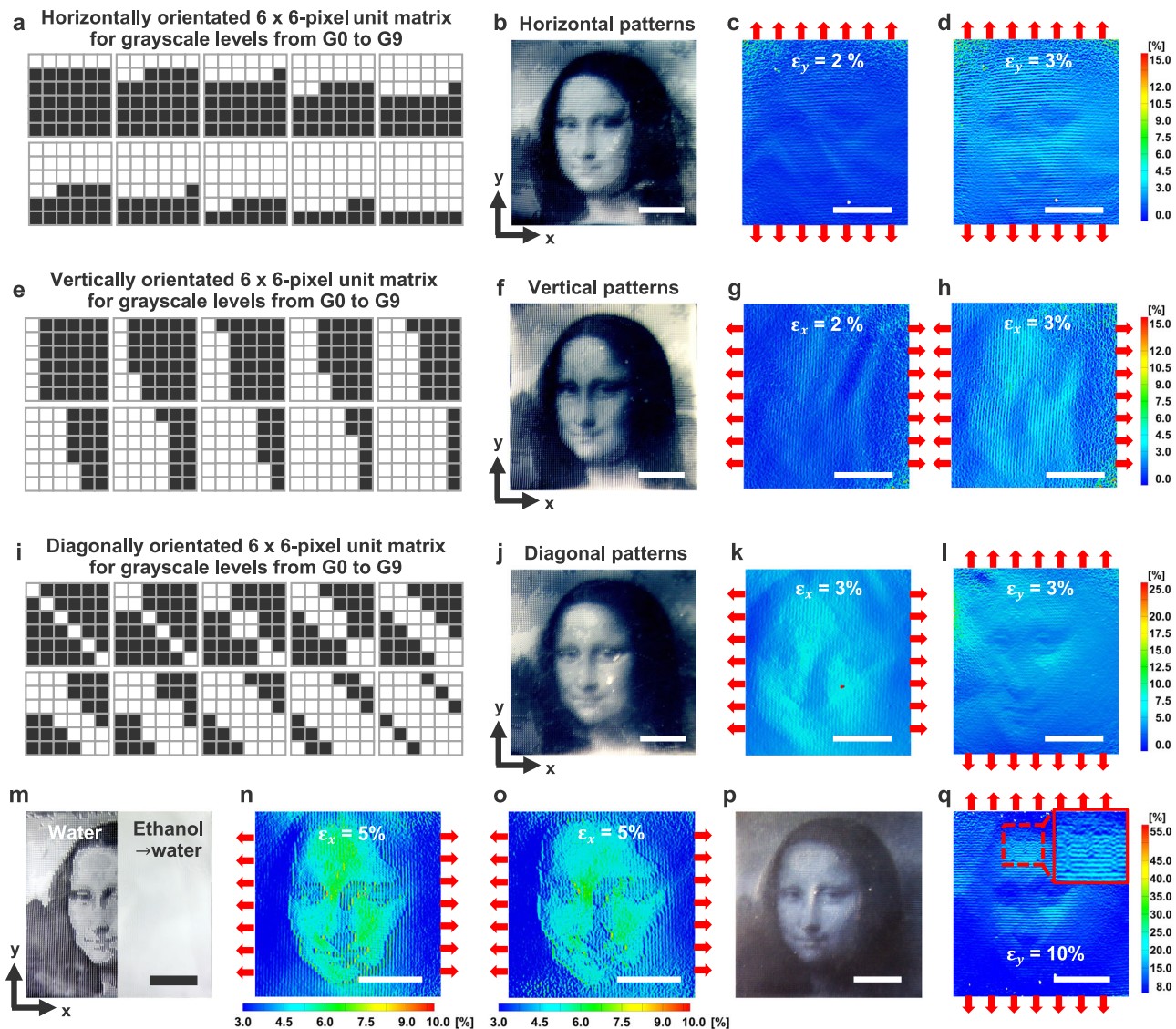

**Fig. 5 | Halftone pattern-regulated strain mapping for multi-modal, multi-layered information encryption-decryption.** Halftone patterns oriented horizontally (**a**), vertically (**e**), or diagonally (**i**) across grayscale levels (G0 to G9), with corresponding optical halftone images of the Mona Lisa (**b**, **f**, **j**) and full-field strain mappings: y-strain (**c**, **d**, **l**) and x-strain (**g**, **h**, **k**) under uniaxial stretching (red arrows). Optical (**m**) and x-strain mapping images (**n**, **o**) of the Mona Lisa encoded with hybrid halftone patterns for enhanced contrast. The left side of the optical image in (**m**) and its x-strain mapping image (**n**) show the deswollen hydrogel in water at 35 °C, while the right side, where the image information is erased in (**m**) but preserved in the x-strain mapping (**o**), represents the film after ethanol immersion followed by water at 35 °C. **p**, **q** Second-layer encryption of the letter "M", composed of diagonally oriented halftone patterns embedded within the horizontally oriented halftone patterns of the Mona Lisa at the same grayscale level, displayed in the optical image (**p**) and the y-strain mapping image under 10% y-axis strain $\varepsilon_y$ (**q**). Scale bars, 5 mm.

For shape-morphed axisymmetric non-Euclidean 3D structures such as spherical caps and hyperbolic saddles, we demonstrated distinct Gaussian curvatures: spherical caps exhibited positive Gaussian curvature (Supplementary Fig. 13b), whereas hyperbolic saddles displayed negative Gaussian curvature (Supplementary Fig. 13d). These curvatures were governed by growth functions for caps (1) and saddles (2) below[28]:

$$\eta = \frac{A_{35\,°C}}{A_0} = \frac{c}{\left(1 + \frac{r^2}{R}\right)^2} \tag{1}$$

$$\eta = \frac{A_{35\,°C}}{A_0} = \frac{c}{\left(1 - \frac{r^2}{R}\right)^2} \tag{2}$$

Following the thermal process Route-1, as the temperature increased from room temperature to 35 °C, the thermo-responsive hydrogel "skins" morphed into their prescribed non-Euclidean 3D shapes, featuring halftone-pattern-regulated Gaussian curvatures and surface textures (Supplementary Fig. 13). The primary shape transition occurred near LCST and continued until 35 °C. Additional biomimetic 3D shapes are shown in Supplementary Fig. 14. These findings demonstrate how 4D-printed dynamic halftone patterns enable spatial control of in-plane growth, facilitating the transformation of 2D hydrogel films into prescribed non-Euclidean 3D structures[28,44,45].

Building on the stimulus-reconfigurable optical and shape-morphing behaviors of 4D-printed hydrogels, we further explored the integration of multifunctionality within a single hydrogel "skin". By co-designing halftone patterns to simultaneously encode letter/image information and growth functions, we enabled the controlled emergence of encoded information as 2D films transformed into prescribed 3D caps (Fig. 6e–h). Similar to soft-bodied cephalopods, these smart "skins", encoded with stimulus-reconfigurable binary halftone

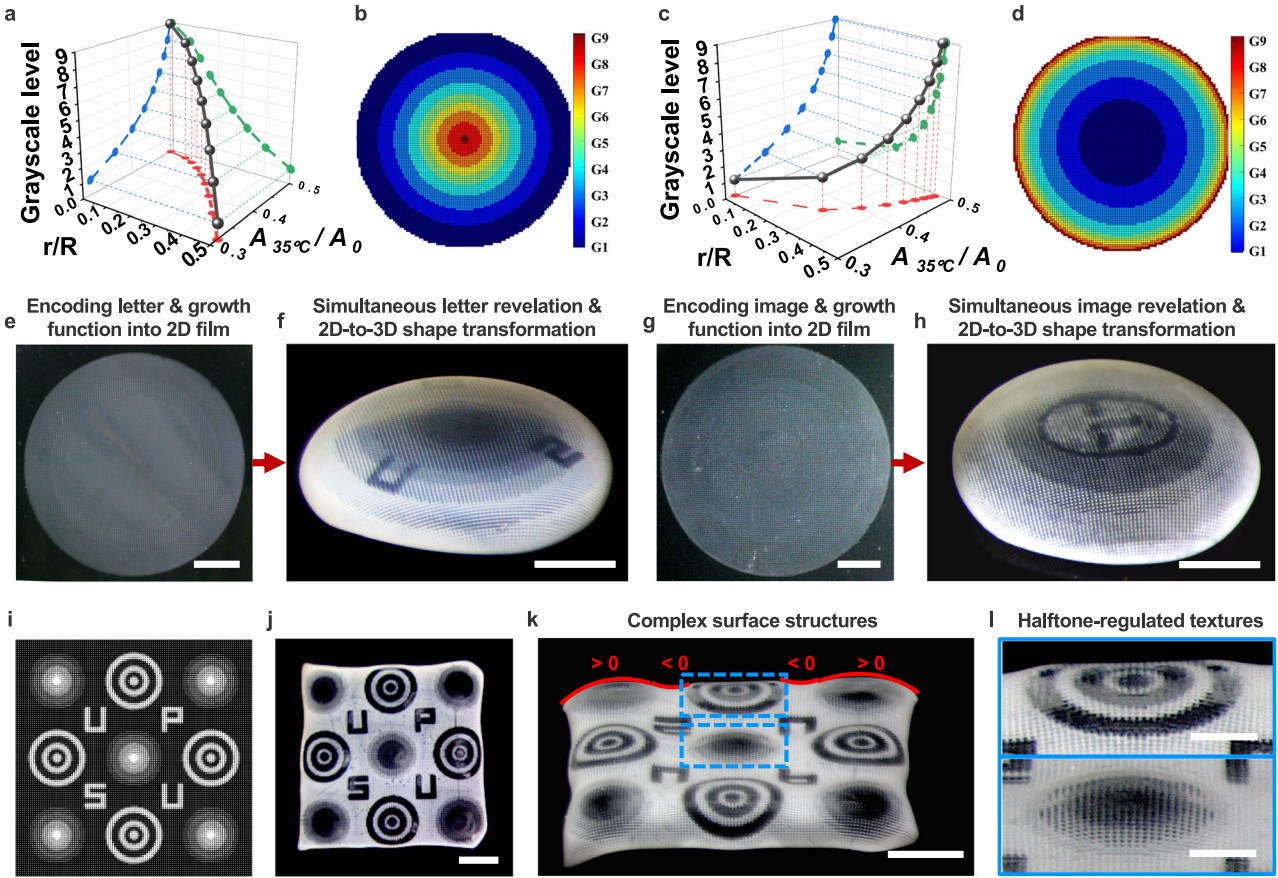

**Fig. 6 | Simultaneous control over stimulus-reconfigurable optical appearance, shape transformation, and surface texture.** Halftone pattern-encoded growth functions enabling the transformation of 2D hydrogel films into axisymmetric non-Euclidean 3D shapes, including spherical caps (**a**, **b**) and hyperbolic saddles (**c**, **d**). The design principles are established through the relationships among areal deswelling ratios ($A_{35°C}/A_O$), grayscale levels (G1-G9), and the relative radius of concentric rings (r/R) (**a**, **c**). Halftone-regulated grayscale images in (**b**, **d**) were produced using MathWorks MATLAB software. **e**–**h** Co-designed halftone patterns simultaneously regulate optical information and growth functions, revealing encoded letters and graphics as the 2D film morphs into prescribed 3D caps. **i**–**l** Shape-morphing of complex surface morphologies featuring varying local Gaussian curvatures (positive, negative, or hybrid) and halftone pattern-regulated textures. Scale bars, 10 mm (**e**, **g**), 5 mm (**f**, **h**, **j**, **k**), 2 mm (**l**).

domains, allow for simultaneous manipulation of optical appearance and global 2D-to-3D shape transformation.

To further mimic the complex surface structures of cephalopods, we integrated diverse halftone arrangements to create hierarchical surface morphologies through deswelling-induced deformation (Fig. 6i–l). For example, small bumpy "papillae" with positive Gaussian curvature emerged on a larger curved surface after deswelling (Fig. 6k). By encoding different halftone-regulated grayscale designs, we achieved localized surface structures with positive, negative, or hybrid Gaussian curvatures, each exhibiting distinct textures (Fig. 6k, l)[7]. This halftone-enabled 4D printing approach offers a versatile strategy for simultaneous control of optical appearance, shape transformation, and surface texture within a single synthetic material, expanding possibilities for dynamic 3D encryption-decryption.

## Discussion

This work introduces a versatile halftone-encoded 4D printing platform capable of digitally programming multifunctional properties—optical appearance, mechanical stiffness, and shape-morphing behavior—within a single synthetic hydrogel system. Inspired by the dynamic body patterning of cephalopods, we developed a strategy to fabricate bioinspired artificial "skins" that respond to environmental stimuli such as temperature, solvents, and mechanical stress. Through binary domain encoding—"0" for soft, lightly crosslinked regions and "1" for stiff, highly crosslinked regions—our method enables spatially controlled co-design of disparate physical properties without the need for multiple inks or multimaterial switching during the printing process.

The key feature lies in the ability to simultaneously couple and decouple mechanical, optical, and shape-morphing features within a monolithic construct. Unlike conventional multimaterial approaches, our single-resin printing technique produces locally distinct behaviors, including sharp contrasts in transparency (opaque vs. transparent), stiffness (40 kPa vs. 123 kPa), and shrinkage (0.3 vs. 0.5). These differences are leveraged to create multifunctional materials capable of dynamic behavior and information encryption.

Specifically, halftone-encoded domains enable the co-design of optical and mechanical properties, where cononsolvency-induced pore formation allows optical modulation, while mechanical heterogeneity enables strain-based imaging. Full-field DIC strain mapping further revealed that, even after visual erasure, encrypted information can be retained within strain patterns under small external loads. Furthermore, the co-design of optical and shape-morphing functionalities was demonstrated through temperature-triggered transformations, where halftone contrast encodes 2D visual patterns while global shrinkage gradients drive 3D shape changes within a single-layer material system. Looking forward, we envision a two-step printing strategy for fabricating a bilayer hydrogel system, in which an artificial skin layer is introduced to decouple functions: the top layer dedicated to visual information and the bottom layer responsible for shape transformation.

While the presented system showcases strong multifunctionality, several limitations remain. The PNIPAm-based responsiveness restricts actuation to aqueous or solvent-rich environments, limiting application flexibility. Future work will explore alternative responsive materials to expand the range of operating conditions. Additionally, the current system produces only grayscale outputs; realizing full-color tunability—critical for more complete emulation of biological camouflage—will require integration of chromophores, photonic crystals, or structural coloration strategies, which may introduce manufacturing complexities. Finally, to further enhance responsiveness, future work will investigate hierarchically porous hydrogel architectures with interconnected networks[31]. Such designs have shown promise in simultaneously accelerating solvent diffusion and enabling dynamic optical-mechanical modulation—addressing current limitations in actuation speed and functional versatility.

Importantly, our halftone-encoded strategy enables dynamic modulation of local binary domains that collectively govern the macroscopic behavior of the material. This binary logic offers a powerful and scalable platform for programming multifunctional responses in soft matter systems. Beyond hydrogels, the method is broadly compatible with other stimuli-responsive materials, including liquid crystal elastomers (LCEs) and shape memory polymers (SMPs), further expanding its applicability to areas such as soft robotics, flexible displays, optical sensing, smart actuators, biomedical devices, and secure communication technologies.

In conclusion, our halftone-enabled 4D printing method offers a robust, scalable strategy for engineering multifunctional synthetic materials with spatially programmable and stimulus-reconfigurable behaviors. By unifying digital design, localized binary encoding, and environmentally triggered actuation, this platform provides a foundation for next-generation soft matter systems with broad implications for stimulus-responsive systems, biomimetic engineering, and advanced encryption technologies.

## Methods

### Materials
Monomer N-Isopropylacrylamide (NIPAm, ≥97%), photoinitiator Phenylbis(2,4,6-trimethylbenzoyl) phosphine oxide (PBPO, ≥97%), and short-chain crosslinker N, N'-Methylenebis(acrylamide) (BIS, ≥97%) were obtained from Fisher Scientific. Long-chain crosslinker Poly(ethylene glycol) diacrylate (PEGDA, Mn 700) was purchased from Sigma-Aldrich. Acetone (≥99.5%) was obtained from VWR, and Amine Functionalized Graphene Oxide Powder (A-GO, ~15μm) was obtained from MSE Supplies (Arizona, US). All chemicals were used as received.

### Preparation of resin solutions
NIPAm precursor resins were prepared by dissolving NIPAm (4 g), PEGDA (1.25 mol% of NIPAm), BIS (1 mol% of NIPAm), and PBPO (0.15 mol% of NIPAm) in a 10 mL aqueous solution with a water-to-acetone volume ratio of 1:3. PBPO was first completely dissolved into acetone before mixing with the resin solution. For specimens used in DIC analysis, A-GO (0.5 mg mL$^{-1}$) was added to the precursor resin. The resulting resins were ultrasonicated until homogeneous and stored in amber glass bottles. Before use, resins were ultrasonicated for 15 minutes and purged with nitrogen for 30 seconds.

### Printing of halftone hydrogel films
Digital halftone stereolithography was performed using an Anycubic Photon D2 digital light processing (DLP) 3D printer (UV light intensity: 2.5 mW cm$^{-2}$, wavelength: 405 nm). The projection cell was built by placing a polydimethylsiloxane (PDMS) spacer with a thickness of 450 - 550 μm on a Teflon FEP film. After purging the resin solution with nitrogen, it was introduced into the cell and covered with a glass coverslip (Brain Research Laboratories, MA). This method utilizes a dynamic mask, generated by the Digital Micromirror Device (DMD)

of the DLP 3D printer (resolution: ~50 μm/pixel), to encode halftone patterns for spatiotemporal control of photopolymerization. To create continuous-tone imaging, different halftoning algorithms (e.g., Fig. 3a, b, and Fig. 5a, e, i) were implemented to generate 10 grayscale levels within a 6 × 6-pixel unit matrix. As an illustrative example, the Mona Lisa image (7000 × 7000-pixel RGB, Fig. 3c) was first converted to a 120 × 120-pixel grayscale image with 256 grayscale levels (Fig. 3d). These grayscale values were then uniformly binned into ten intervals, corresponding to the grayscale levels represented by predefined halftone patterns labeled G0 to G9. The resulting image was transformed into a 720 × 720-pixel halftone pattern and processed in MATLAB to generate stereolithography (STL) files. Notably, the varying heights of black and white pixels in the STL files of the halftone designs correspond to the controlled exposure levels. Specifically, black pixels with lower heights represent 50-second exposure, while white pixels with greater heights represent 120-second exposure (Supplementary Fig. 15). These STL files were subsequently sliced into 2D projection layers using Anycubic Photon Workshop software for DLP printing. In addition to DLP 3D printer, LCD based 3D printer (Anycubic Photon Mono) was also used to encode halftone pattern-regulated growth function, demonstrating 2D-to-3D shape transformations (e.g., Supplementary Figs. 13, 14). After fabrication, the hydrogel films were detached from the glass coverslip, rinsed with ethanol to remove unreacted chemicals, and swollen in deionized (DI) water at 4 °C for 12 h, with water changes every 6 h.

### Preparation of hydrogels films for morphological characterizations
To investigate the relationship between the optical appearance of hydrogel films and their microstructural morphologies, various hydrogel samples were prepared, including highly crosslinked, lightly crosslinked, and halftone patterned hydrogels. These films were processed under different states and solvent solutions: (1) the swollen state at room temperature in water (25 °C); (2) the deswollen state in water at 35 °C; (3) as-printed hydrogel films washed with ethanol for 30 seconds (retaining an ethanol-water mixture within the hydrogel network); (4) deswollen hydrogel films immersed in ethanol for 15 seconds, followed by water at 35°C. All hydrogel films were then shock-frozen in liquid nitrogen, fractured using a blade, and lyophilized at -105 °C for 3 days using a benchtop freeze-dry system (FreeZone 2.5 L, Labconco Corp.).

### Material characterization of hydrogel films
Hydrogel samples for SEM characterization were sputter-coated with a 5 nm thick iridium using an EM ACE600 sputter coater (Leica Microsystems). Microstructural characterization of the hydrogels in different states and under varying conditions was conducted using a SEM at an accelerating voltage of 10 kV (Verios G4, Thermo Scientific). Optical images were captured with a Trinocular Stand Stereo Zoom Microscope (AmScope), and surface areas were analyzed using ImageJ software (NIH, USA). Movies were captured using a Dino-Lite Edge digital microscope. Hydrogel films encoded with designed halftone patterns (9.18 mm × 9.18 mm) were used to investigate the deswelling behaviors. The areas of the hydrogel films in their deswollen state at 35 °C (denoted as $A_{35°C}$) were measured by gradually heating the samples from 25 to 35 °C. The areal deswelling ratios were calculated as $A_{35°C}/A_0$, where $A_0$ represents the area of the hydrogel film in its as-printed state.

### Mechanical characterization of hydrogel films
The mechanical properties of hydrogel films were evaluated using a horizontal motorized UniVert mechanical tester equipped with a 2.5 N load cell (CellScale, Canada). All tests were conducted in a water bath maintained at 35 °C. Tensile tests were performed with a gauge length

of 20 mm and a stretching speed of 10 mm min$^{-1}$, with data collected at a frequency of 20 Hz. Sample thickness was measured using a micrometer caliper. Young's modulus was determined from the initial slope of the stress-strain curve within the 1–20% strain range.

## In-situ strain mapping via DIC analysis

In-situ strain mapping was performed using an ARAMIS High-Speed DIC System (Zeiss GOM Correlate, Trilion, PA, USA). Full-field surface strain mapping images were captured by a front-facing camera during external loading, with a minimum analytical resolution of 25 μm. The DIC system was configured to record images at a rate of one frame per 1 % strain. Time-resolved 2D grayscale images were processed to generate real-time, full-field strain maps. Hydrogel samples for DIC analyses were prepared by mixing A-GO marker with the resin.

## Data availability

The data supporting this study's findings are available in the Article and its Supplementary Information, or from the corresponding author upon request. The source data generated in this study are provided in the Source Data file. Source data are provided with this paper.

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

## Acknowledgements

H.S. acknowledges financial support from the Pennsylvania State University start-up fund.

## Author contributions

H.S. conceived the research. H.S. and H.Y. designed the experiments. H.Y. conducted the design, printing, and characterization of hydrogels. H.L. and J.Z. contributed to hydrogel printing and characterization. H.J.Q. and T.L. contributed to discussions and data analysis. H.S. and H.J.Q. supervised the research. H.S., H.Y., T.L., and H.J.Q. co-wrote the manuscript with input from all authors. All authors discussed the results and commented on the manuscript.

## Competing interests

H.S. and H.Y. are inventors on patents (US Provisional Application No. 63/840,228) relating to this study filed by the Pennsylvania State University, University Park. H.S. and H.Y. have no other competing interests. H.J.Q., T.L., H.L., and J.Z. declare no competing interests.
