## [Transparent Peer Review file · Nature Communications]

Halftone-Encoded 4D Printing of Stimulus-Reconfigurable Binary Domains for Cephalopod-Inspired Synthetic Smart Skins

Corresponding Author: Professor Hongtao Sun

Version 0:

Reviewer comments:

Reviewer #1

(Remarks to the Author)

In this work, the authors proposed an interesting phoneme in which two apparently similar hydrogels with different cross-linking degrees can exhibit distinct transitions under external stimuli. Furthermore, they demonstrate a dynamic graphical display for information encryption. However, they fail to explore the underlying mechanism and instead focus more on the graphical display. In fact, the concepts of transition changes, digital light 4D printing (Nat. Commun. 2018, 9, 3705), shape-morphing (Nat. Commun. 2021, 12, 603), and information encryption (Nat. Commun. 2025, 16, 2830) are not entirely unprecedented. This somewhat diminishes the novelty of the approach. It is hard to persuade me that this information encryption approach offers significant advantages compared to existing pattern-dependent or morphology-based encryption methods, which should be discussed in more detail. Unless the authors can provide a thorough explanation of the scientific mechanisms involved, the current manuscript is not suitable for publication in Nature Communications.

1. The role of cross-linkers should be clarified, specifically why PEGDA and BIS are used together.
2. The critical UV exposure time at which the two distinct transition events occur should be identified.
3. The cosolvency effect is well-known in PNIPAm hydrogels. What about using DMSO or DMF as cosolvents.
4. The degree of cross-linking can influence the LCST of the PNIPAm network. Does the phoneme suggest that highly crosslinked hydrogels become transparent due to the increase in LCST? The existing explanation is confusing.
5. What is the trend in transparency change at different grayscale levels?
6. How can the flatness of the swollen hydrogel be controlled due to the patterned cross-linking distribution?
7. The presence of cross-linking distribution would induce 3D morphologies. How does it influence the 2D display, or how can it be balanced?

Reviewer #2

(Remarks to the Author)

In this manuscript, Sun et al. reported a 4D printing strategy that imparts programmable control over optical appearance, mechanical properties, surface texture, and shape transformation within a single smart hydrogel film in response to various external stimuli. This approach shows potential for certain applications such as smart skins and soft robots. I recommend acceptance of this manuscript after careful revisions. Below are some comments and questions that need clarification:

Q1. In the abstract, the authors mentioned that the 4D printing method enables simultaneous and programmable control over optical appearance, mechanical properties, surface texture, and shape transformation within a single smart hydrogel film. However, the manuscript lacks conclusive experimental evidence demonstrating the spatiotemporal coupling of optical, mechanical, shape-morphing, and textural responses under identical stimuli, as claimed. How to achieve programmable control over multiple properties at the same time?

Q2. In the introduction, the authors claimed capabilities beyond existing synthetic materials. Authors should provide a comparative evaluation to highlight the contributions of this work, helping readers understand its significance.

Q3. In the text (line 48-49, page 3), it is written that "However, achieving refined, multi-faceted control over property and shape changes in synthetic materials remains a significant challenge...".

I think a deeper summary of the challenges in this field is necessary, as it will help readers understand the innovations of this work.

Q4. It is high effective to use a 4D printing technique to achieve precise spatiotemporal control over photopolymerization

within smart hydrogel films. However, further details on how the experimental procedure was devised are required, and the authors should provide a more comprehensive description of the methodology.

Q5. The rationale for selecting UV exposure times requires further clarification. The choice of only two exposure times (50 s and 120 s) appears arbitrary. Expanding the tested range (e.g., 80 s, 100 s, and 140 s) would strengthen the structure-property correlation between exposure times and material performance.

Minor points:

Q1. Figure 1: Some of the scale bars are difficult to make out from the background color. Please use different colors to improve the contrast.

Q2. Supplementary Fig 7: The unit format appears inconsistent: strain (%) is used in Fig. S7a, whereas strain is shown without units in Figs. S7b, c and Fig. 4. Please use a consistent format.

Reviewer #3

(Remarks to the Author)

The current paper introduces a halftone-encoded 4D printing method that enables simultaneous and programmable control over optical appearance, mechanical properties, surface texture, and shape transformation within a single smart hydrogel film in response to external stimuli, including temperature, solvents, and mechanical stress. The resolution and versatility of the proposed method are impressive. The reported data and results are solid. Publication of the paper in NC is recommended. The authors are suggested to consider the following comments to further strengthen the paper:

1. The authors may want to discuss the timescales for transitions under different conditions. Additionally, the authors may discuss potential future works when fast-transition applications are required.
2. What is the minimum feature size (pixel) at which halftone fidelity breaks down due to reasons such as radical diffusion or overcuring?
3. Is there a tradeoff between resolution and the optical/mechanical contrast achieved?
4. How many stimulus cycles can the material undergo before losing contrast?
5. How do ambient environments impact hydrogel function? Are there encapsulation strategies that could extend lifetimes?

Version 1:

Reviewer comments:

Reviewer #1

(Remarks to the Author)

The current version after revision could be accepted as is.

Reviewer #2

(Remarks to the Author)

During the revisions, the authors have generally addressed my concerns. I think the scientific story is completed with impressing results. An acceptance for publication of this work is recommended.

Reviewer #3

(Remarks to the Author)

Prompt publication on NC is recommended.

Point-to-Point Response to Reviewers

Reviewer #1 (Remarks to the Author):

In this work, the authors proposed an interesting phoneme in which two apparently similar hydrogels with different cross-linking degrees can exhibit distinct transitions under external stimuli. Furthermore, they demonstrate a dynamic graphical display for information encryption. However, they fail to explore the underlying mechanism and instead focus more on the graphical display. In fact, the concepts of transition changes, digital light 4D printing (Nat. Commun. 2018, 9, 3705), shape-morphing (Nat. Commun. 2021, 12, 603), and information encryption (Nat. Commun. 2025, 16, 2830) are not entirely unprecedented. This somewhat diminishes the novelty of the approach. It is hard to persuade me that this information encryption approach offers significant advantages compared to existing pattern-dependent or morphology-based encryption methods, which should be discussed in more detail. Unless the authors can provide a thorough explanation of the scientific mechanisms involved, the current manuscript is not suitable for publication in Nature Communications.

Response: We sincerely thank the reviewer for carefully reading our manuscript and for the constructive feedback, which have motivated us to further strengthen our manuscript.

We agree with the reviewer that the concepts of transition changes, 4D printing, shape-morphing, and information encryption have been explored in prior studies. However, our work introduces a fundamentally new strategy: a halftone-encoded printing method that digitally programs multifunctional material properties by encoding binary domains—“0” (lightly crosslinked) and “1” (highly crosslinked)—within a single precursor resin system. This approach eliminates the need for multiple inks or multimaterial switching and allows co-design of mechanical, optical, and shape-morphing behaviors with spatial and functional precision (Fig. R1). To the best of our knowledge, such a digitally encoded, single-material-based co-design strategy has not been previously reported, and its underlying mechanisms remain unexplored in the current literature.

In our system (Fig. R1), these binary domains demonstrate significantly different physical properties: for example, Young's modulus (40 kPa vs. 123 kPa), optical appearance (opaque vs. transparent), and shrinkage ratio (0.3 vs. 0.5). By spatially arranging these domains and their integration, we can program the local and global multiphysical behaviors of the printed construct (Fig. R1a). More importantly, this strategy enables a unique halftone-encoded 4D printing platform for co-designing multifunctionality within a monolithic hydrogel system. Inspired by cephalopods' dynamic body patterning, we built a general design and printing framework that allows simultaneous tuning of optical (Fig. R1a top panel), mechanical (Fig. R1b-c), and swelling/deswelling ratios governing shape-morphing behaviors (Fig. R1d). The following section elaborates on our co-design strategy for multifunctional controls.

Figure R1. Dynamic halftone-encoded films to program multiphysical behaviors. **a**, Spatial binary halftone-encoded printing method enables the co-design of optical, mechanical and shape-morphing behaviors. **b**, Stress-strain curves of hydrogel films encoded with FM halftone patterns. **c**, The comparison among Young's moduli. **d**, Swelling/deswelling ratio of hydrogel films encoded with FM and AM halftone patterns under different temperatures.

Co-design of optical and mechanical features. Our halftone-encoded approach enables simultaneous and spatially resolved control over local mechanical and optical properties. The cononsolvency-induced phase separation and pore formation in the surface skin layers enable dynamic modulation of optical contrast, while the mechanical heterogeneity introduced via halftone encoding allows for tunable strain distributions, which we visualized through full-field DIC strain mapping (Figs. R2). For instance, after immersing the printed hydrogel in ethanol followed by water at 35 °C, the optical information was erased due to solvent-induced opacity change (Fig. R2n). Yet, strain mapping retained a clear mechanical image of the Mona Lisa (Figs. R2 o-p), confirming that our approach supports multi-modal encryption and decryption based on orthogonal physical properties.

Figure R2. Co-design of optical and mechanical features via DIC strain-mapping analysis. a-c, Optical images and x-strain mapping of deswollen hydrogel films (35 °C water) encoded with G3 (grayscale level) half-tone patterns oriented at 0° (a), 45° (b), and 90° (c) under 30% strain. d, Stress-strain curves of deswollen hydrogel films corresponding to (a-c) showing anisotropic mechanical behaviors. e-k, Half-tone patterns oriented horizontally, vertically, or diagonally across grayscale levels (G0 to G9), with corresponding optical half-tone images of the Mona Lisa (e, h, k) and full-field strain mappings: y-strain (f, i, l) and x-strain (g, j, m) under uniaxial stretching (red arrows). n-p, The left side of the optical image in (n) and its x-strain mapping image (o) show the deswollen hydrogel in water at 35°C, while the right side, where the image information is erased in (n) but preserved in the x-strain mapping (p), represents the film after ethanol immersion followed by water at 35°C.

Co-design of optical and shape-morphing features. By exploiting temperature-responsive shape transformations and optical contrast modulation, we demonstrate that encoded information can dynamically appear or disappear as 2D films transform into prescribed 3D structures (Figs. R3a-d) with controlled textures (Figs. R3f-k). Importantly,

local halftone contrast for information encoding and the global shrinkage gradient responsible for shape-morphing can be concurrently occurred in our current single layer hydrogel system. In addition, surface skin layers emerge when the hydrogel is immersed in different solvents (Fig. R3h–k), highlighting the potential to simultaneously regulate optical properties (via the skin layer) and deformation behaviors (via the underlying layer). In future work, we plan to develop a two-step printing process to introduce an artificial skin layer that fully decouples the two functions: a thin top layer dedicated to image display and revelation, and a secondary layer responsible for shape transformation (Fig. R3l).

Figure R3. Co-design of optical and shape-morphing features. a-c, Co-designed halftone patterns simultaneously regulate optical information and growth functions, the revealed letters and graphics in swollen flat hydrogel films (a, d) were retrieved as the 2D film morphs into prescribed 3D caps (b, d). e, Optical appearance of hydrogel films printed with low (50-s, “0”) and high (120-s, “1”) UV exposures in the deswollen state at 35 °C water when placed against a black background. f-g, Surface views (f) and cross-sectional views (g) of hydrogel films in their deswollen state reveal textures control. h-k, The cross-sectional view (h, j) and top view (i, k) of deswollen hydrogel films after immersion in ethanol for 15 seconds, followed by water immersion at 35°C. l, A strategy that decouple optical appearance and shape transformations by using two-layer design, top layer is manipulated with halftone-encoded patterns for optical behaviors, and bottom layer is taken for 2D-to-3D shape transformations.

Co-design of mechanical and shape-morphing features. In our prior study (Nat. Commun. 2024, 15, 9268), we have demonstrated simultaneous control over mechanical properties and shape transformation, along with their decoupling effect by incorporating cellular-like unit structures (e.g., Figs. R4f–i). In the current work, our halftone-encoded printing approach also enables the fabrication of cellular-like units (Figs. R4a–e). This design strategy enables simultaneous tailoring of stiffness and programmable deformation. As this work primarily focuses on optical properties and their coupling or decoupling with other functionalities, the co-design of mechanical and shape-morphing features is not included in the current study.

Moreover, our halftone-encoded printing method is compatible with various smart material systems, including liquid crystal elastomers (LCEs) and shape memory polymers (SMPs). This versatility significantly broadens the potential applications of our method in soft robotics, sensing, information displays, and biomedical devices.

To better highlight the novelty and underlying scientific mechanisms of this co-design platform, we have substantially expanded the mechanistic discussions in the revised manuscript and added more discussions summarizing these key novelties and future perspectives. The revised discussion and conclusion section is appended below.

Figure R4. Co-design of mechanical and shape-morphing features. a–d, Stiff re-entrant honeycomb meta-structure (G9 halftone) embedded in a soft film (G1 halftone), demonstrate co-designed optical and mechanical behaviors in its deswollen state with corresponding optical halftone images (b) and full-field strain mappings: y-strain (c) and x-strain (d) mappings under y-axis loading under 20% y-axis strain. e, Strain profiles along a marked linear pathway X. f–i, Cellular units enable the simultaneous regulation of mechanical and shape transformations (Nat. Commun. 2024, 15, 9268). Precise control over the curvatures and angles in 3D caps (f) and hyperbolic saddles (h) alongside the corresponding growth functions dictating the shape transformations (g, i), The L in (g and i) represents the variable arc length on the tunable surface of the shape morphed cap.

Discussion and Conclusion

This work introduces a versatile halftone-encoded 4D printing platform capable of digitally programming multifunctional properties—optical appearance, mechanical stiffness, and shape-morphing behavior—within a single synthetic hydrogel system. Inspired by the dynamic body patterning of cephalopods, we developed a strategy to fabricate bioinspired artificial “skins” that respond to environmental stimuli such as temperature, solvents, and mechanical stress. Through binary domain encoding—“0” for soft, lightly crosslinked regions and “1” for stiff, highly crosslinked regions—our method enables spatially controlled co-design of disparate physical properties without the need for multiple inks or multimaterial switching.

The key innovation lies in the ability to simultaneously couple and decouple mechanical, optical, and shape-morphing features within a monolithic construct. Unlike conventional multimaterial approaches, our single-resin printing technique produces locally distinct behaviors, including sharp contrasts in transparency (opaque vs. transparent), stiffness (40 kPa vs. 123 kPa), and shrinkage (0.3 vs. 0.5). These differences are leveraged to create multifunctional materials capable of dynamic behavior and information encryption.

Specifically, halftone-encoded domains enable the co-design of optical and mechanical properties, where cononsolvency-induced pore formation allows optical modulation, while mechanical heterogeneity enables strain-based imaging. Full-field DIC strain mapping further revealed that, even after visual erasure, encrypted information can be retained within strain patterns under small external loads. Furthermore, the co-design of optical and shape-morphing functionalities was demonstrated through temperature-triggered transformations, where halftone contrast encodes 2D visual patterns while global shrinkage gradients drive 3D shape changes within a single-layer material system. Looking forward, we envision a two-step printing strategy for fabricating a bilayer hydrogel system, in which an artificial skin layer is introduced to decouple functions: the top layer dedicated to visual information and the bottom layer responsible for shape transformation.

While the presented system showcases strong multifunctionality, several limitations remain. The PNIPAm-based responsiveness restricts actuation to aqueous or solvent-rich environments, limiting application flexibility. Future work will explore alternative responsive materials to expand the range of operating conditions. Additionally, the current system produces only grayscale outputs; realizing full-color tunability—critical for more complete emulation of biological camouflage—will require integration of chromophores, photonic crystals, or structural coloration strategies, which may introduce new manufacturing complexities. Finally, to further enhance responsiveness, future work will investigate hierarchically porous hydrogel architectures with interconnected networks. Such designs have shown promise in simultaneously accelerating solvent diffusion and enabling dynamic optical-mechanical modulation—addressing current limitations in actuation speed and functional versatility.

Importantly, our halftone-encoded strategy enables dynamic modulation of local binary domains that collectively govern the macroscopic behavior of the material. This binary logic offers a powerful and scalable platform for programming multifunctional responses in soft matter systems. Beyond hydrogels, the method is broadly compatible with other stimuli-responsive materials, including liquid crystal elastomers (LCEs) and shape

memory polymers (SMPs), further expanding its applicability to areas such as soft robotics, flexible displays, optical sensing, smart actuators, biomedical devices, and secure communication technologies.

In conclusion, our halftone-enabled 4D printing method offers a robust, scalable strategy for engineering multifunctional synthetic materials with spatially programmable and stimulus-reconfigurable behaviors. By unifying digital design, localized binary encoding, and environmentally triggered actuation, this platform provides a foundation for next-generation soft matter systems with broad implications for stimulus-responsive systems, biomimetic engineering, and advanced encryption technologies.

Comment 1. The role of cross-linkers should be clarified, specifically why PEGDA and BIS are used together.

Response: We thank the reviewer for this comment. The PEGDA and BIS crosslink the polymer at the same time, but at different reaction rates. We refer to the propagation reaction rate R_p in Ref. 40, $R_p = k_p [M][Mn \bullet]$, where k_p represents the propagation rate constant, $[M]$ represents the double bond concentration, and $[Mn \bullet]$ represents the total propagating radical concentration. Considering PEGDA and BIS both contain two double bonds at their ends and the concentration of PEGDA is higher than that of BIS, $[M_{c=c \text{ from PEGDA}}] > [M_{c=c \text{ from BIS}}]$, thus PEGDA dominates the earlier polymerization. However, with the increased curing time, the further crosslinking process was due to the increased participation of short-chain BIS. More importantly, it has been reported that single crosslinker could not enable the dynamic modification of the mechanical properties^{24, 26, 33}.

To clarify the function of PEGDA and BIS in the material system, we add more details in the main manuscript and also summarize as below.

(Line 105: In this study, we used photocurable poly(N-isopropylacrylamide) (PNIPAm) hydrogels, crosslinked with a combination of a long-chain crosslinker, poly(ethylene glycol) diacrylate (PEGDA), and a short-chain crosslinker, N, N'-Methylenebis(acrylamide) (BIS), in an acetone-water solvent. Our study indicates that the PEGDA and BIS can simultaneously tune the crosslinking properties of the hydrogel film, but to different extents. Under lower curing time (50 seconds), the hydrogel network was primarily crosslinked by the long-chain PEGDA and presented a loose microstructure; while under high curing time (120 seconds), more short-chain BIS participated and dominated the following crosslinking process, and enhanced the network density and strength²⁴⁻²⁶. By varying UV exposure times during printing, we controlled the degree of crosslinking in the hydrogels, resulting in distinct variations in optical transmittance, mechanical properties, and thermally induced growth (expansion and contraction) during the swelling-deswelling cycle. These differences stem from distinct dynamic transitions between intermolecular and intramolecular interactions^{27, 28}.)

Comment 2. The critical UV exposure time at which the two distinct transition events occur should be identified.

Response: We appreciate the reviewer's insightful question regarding the critical UV exposure times. Below, we clarify our rationale for selecting the current exposure times.

Our selection of 50-s and 120-s as key thresholds was guided by the need to balance structural fidelity, functional performance, and reproducibility. At 50-s exposure, the hydrogel reaches sufficient crosslinking density to maintain mechanical stability while exhibiting an opaque appearance (Fig. R5a). Further exposure up to 120-s gradually reduces this opacity, resulting in a transparent state, as confirmed by microscopic imaging. This transition window (50–120 s) was deliberately chosen to avoid undercuring (<50-s), which compromises structural integrity, or overcuring (>120-s), which leads to excessive brittleness (Fig. R5b).

Moreover, the shrinkage ratio—a critical parameter for 2D-to-3D shape transformation—scales predictably with exposure time, ranging from ~ 0.3 ($(A_{35^\circ\text{C}}/A_0)_{50\text{-s}}$ at G0) to ~ 0.5 ($(A_{35^\circ\text{C}}/A_0)_{120\text{-s}}$ at G9) (Fig. R5b). This controllable deformation range allows precise spatial tuning of the hydrogel's in-plane growth during deswelling-induced shrinkage. Importantly, mechanical characterization (Fig. R5c) revealed that properties such as modulus and toughness evolve continuously across this exposure range, with performance peaks aligning with the 50–120 s window. Deviations outside this range disrupt the coupling between optical, mechanical, and shape-morphing behaviors, which are essential for the system's multifunctionality. Thus, these thresholds represent optimized conditions to achieve the desired dynamic responses while ensuring reproducibility.

According to the reviewer's recommendation, the microscopic images, stress-strain curves, and areal swelling/deswelling ratio ($A_{35^\circ\text{C}}/A_0$) of hydrogel films modulated solely by curing times have been added to the Supplementary information (Fig. S2). More explanation has been added to the main manuscript, and is briefly summarized below.

“Line 118: We first examined variations in the optical transmittance of hydrogel films printed with varying UV exposures under different swelling-deswelling states (Supplementary Fig. 1 and 2). ...

Line 134: However, upon heating to 35 °C (above the LCST), the films deswelled, exhibiting distinct optical appearances in their fully deswollen states: lightly crosslinked hydrogels (50-s) turned opaque; opacity gradually diminished with longer curing times (60-s – 110-s); and highly crosslinked hydrogels (120-s) became transparent (iv in Fig. 2a and Supplementary Fig. 2). This progression generated a striking black-and-white visual contrast against a black background (iv in Fig. 2a, Supplementary Figs. 1d and 2).”

“Line 414: Using the FM method to encode grayscale levels from G1 to G9, we achieved areal deswelling ratios $A_{35^\circ\text{C}}/A_0$ ranging from approximately 0.3 to 0.5, as depicted by the projected blue curves (Fig. 6a, c). These halftone-encoded grayscale levels regulated deswelling-induced deformations in a manner similar to those observed under varying curing times (Supplementary Fig. 12).

Figure R5. UV exposure-regulated optical appearance and properties. **a**, Optical views of hydrogel films regulated by UV exposures demonstrate a decrease in opacity with increased curing times (30-s to 140-s) over a black background. **b**, Master curves of curing time-dependent deswelling ratios, showing a range of 0.21 ~ 0.51. **c**, Stress-strain curves of deswollen hydrogel films corresponding to (a).

Comment 3. The cosolvency effect is well-known in PNIPAm hydrogels. What about using DMSO or DMF as cosolvents.

Response: We sincerely thank the reviewer for this comment. The cononsolvency effect has been well studied for multiple water-organic mixtures, such as water-DMSO and water-DMF. The thermodynamic behavior of PNIPAm in water-DMSO or water-DMF solvent is very similar to that of PNIPAm in water-Ethanol solvent²⁹⁻³²; they all show UCST behavior at low water fraction and LCST behavior at high water fraction. However, the specific transition temperatures at different concentrations of organic content vary with

species. For example, in the water-ethanol solvent, the LCST behavior disappears at the mole fraction of Ethanol $x_{Ethanol,LCST} = \sim 15\%$ and the UCST behavior was observed at $x_{Ethanol,UCST} = \sim 28\%$. However, for DMSO and DMF system, these mole fractions change to $x_{DMSO,LCST} = \sim 15\%$, $x_{DMSO,UCST} = \sim 60\%$, and $x_{DMF,LCST} = \sim 26.7\%$, $x_{DMF,UCST} = \sim 28.3\%$. For the LCST behavior, the increase in organic concentration causes a gradual decrease of LCST, which was the contribution of the hydrogen bond formation between water and amide group (-CONH-), and the shielding effect of water on the isopropyl group ((CH₃)₂CH-). However, for the UCST behavior, the higher the organic solvent concentration, the lower the UCST. Considering the polarity of organic molecules, the hydrogen bonds primarily form between water and organic molecules at low temperature; as the temperature increases, polymer-solvent interactions overcome the interchain attractions to redissolve the NIPAm.

According to the reviewer's recommendation, we have made additional experiments using DMSO or DMF as the organic solvent for the information encryption/decryption (Figs. R6-7). Like Ethanol, DMSO and DMF also enable repeatable and reversible information concealment and retrieval. However, these two organic solvents differ from ethanol in the speed and extent of reveal and retrieval.

For instance, when a decrypted halftone-coded film was transferred from ice water to DMSO (Fig. R6a, b), the low-exposed regions rapidly transitioned from opaque to transparent, which is consistent with the phenomenon observed in ethanol, due to the rapid differential of organic solvent into the loosen network ($x_{Ethanol,LCST} = \sim 15\% \approx x_{DMSO,LCST} = \sim 15\%$). However, the high-exposed regions transition more slowly to transparent in DMSO than in ethanol, because DMSO takes longer than ethanol to diffuse into the dense network to reach the concentration required for LCST-UCST transition ($x_{Ethanol,UCST} = \sim 28\% < x_{DMSO,UCST} = \sim 60\%$). Moreover, the DMSO-infiltrated hydrogel film exhibited faster reswelling kinetics when re-immersed in ice water (Fig. R6c) than its ethanol-infiltrated counterpart. This is attributed to DMSO's superior chain-solvating ability and non-protic nature, which maintains the polymer chains in a more expanded state, in contrast to ethanol's protic nature that promotes chain aggregation and strong hydrogen bonding.

Additionally, When DMF was used as a solvent, the ability of hydrogel to encrypt and decrypt information was greatly reduced (Fig. R7), which was due to the small gap between LCST and UCST transition ($x_{DMF,LCST} = \sim 26.7\%$, $x_{DMF,UCST} = \sim 28.3\%$), the two-phase region in the phase diagram was difficult to observe.

Figure R6. A halftone-encoded hydrogel film for image concealment and retrieval through a DMSO-water immersion cycle. a, Real time 0''- 30'': immerse the halftone-encoded hydrogel film in ice water. **b, Real time 30''- 2'30'':** transfer the hydrogel into DMSO. **c, Real time 2'30''- 5'50'':** re-immerses the hydrogel into ice water.

Figure R7. A halftone-encoded hydrogel film for image concealment and retrieval through a DMF-water immersion cycle. a, Real time 0''- 30'': immerse the halftone-encoded hydrogel film in ice water. **b,**

Real time 30''- 60'': transfer the hydrogel into DMF. **c**, Real time 60''- 1'30'': re-immerses the hydrogel into ice water. **d**, Real time 1'30'' – 2'10'': re-immerses the hydrogel into DMF. **e**, Real time 2'10'' – 2'50'': re-immerses the hydrogel into DMF.

Comment 4. The degree of cross-linking can influence the LCST of the PNIPAm network. Does the phoneme suggest that highly crosslinked hydrogels become transparent due to the increase in LCST? The existing explanation is confusing.

Response: We sincerely thank the reviewer for this comment. We agree with the reviewer that the degree of crosslinking can influence the LCST of the PNIPAm network. However, the transition temperature in this material system was observed to occur at 32°C ~ 33 °C, which is below the selected measuring temperature of 35 °C, and all hydrogel films are in their fully-deswollen states.

Previous studies have demonstrated that PNIPAm hydrogels typically become opaque and white color above the LCST due to light scattering induced by microphase separation^{24, 35, 37, 38}. The optical transmittance is strongly influenced by fabrication conditions, including crosslinker type, crosslinking density, and microporous structure. In our material system, hydrogel films exhibited a progressive reduction in opacity as exposure time increased from 50-s to 120-s (Fig. R5a), which correlated with distinct microstructural changes observed in SEM. Additionally, the heating rate further modulated hydrogel transparency (Fig. R8). We attribute these optical variations to the synergistic effects of the PEGDA/BIS dual-crosslinking system on the hydrogel microstructure, as well as thermal processing conditions.

Specifically, shorter UV exposure (50 seconds) generated lightly crosslinked hydrogels with larger pores in the swollen state, enhancing microphase separation between water-rich and polymer-rich regions during deswelling. Extended curing (120 seconds) led to the subsequent incorporation of PEGDA and BIS crosslinkers formed a densely crosslinked network with smaller pores in the swollen state. Upon deswelling, this network collapsed into a dense interior with a relatively smoother surface.

During early-stage crosslinking, long-chain PEGDA formed large and loose connected pores, these macroporous structures collapsed, causing drastic phase separation in the PNIPAm hydrogel and create a substantial refractive index mismatch with the aqueous phase. In the extended curing duration, the subsequent incorporation of short and rigid BIS crosslinkers tightened the network, reducing pore size and restricting polymer chain motion during deswelling. This suppressed large-scale phase separation, minimized refractive index variations, and preserved optical transparency. To address potential concerns, we have expanded the discussion in the manuscript.

4D printed halftone patterns in smart hydrogels for information decryption

Figure R8. 4D-printed halftone-patterned hydrogel films enabling dynamic information decryption. The encrypted image briefly became visible for approximate 10 seconds when placed in ice water after ethanol washing, observed against a black background (orange dot-labeled images); the decrypted graphic information faded as the hydrogel film turned fully transparent when immersed in ethanol (dark red dot-labeled image); the films appeared uniformly opaque and light gray when full swollen in water at 25 °C (green dot-labeled images); graphic information gradually emerges when subjected to two heating protocols across the LCST, with the initial phase separation (brown dot-labeled images) and then in the deswollen state (blue dot-labeled images).

Comment 5. What is the trend in transparency change at different grayscale levels?

Response: We sincerely thank the reviewer for this comment. In the halftone matrix, lower cumulative exposure, represented by darker tones, results in greater opacity in deswollen hydrogel films (e.g., Fig. R1a G0-G2). Conversely, higher cumulative exposure, represented by brighter tones, yields more transparent regions that appear black against a black background (e.g., Fig. R1a G6-G9). For the homogeneous hydrogel films at the deswollen state, they show gradual opacity-to-transparency transition as UV curing time increases from 30-s to 140-s (Fig. R5a).

Comment 6. How can the flatness of the swollen hydrogel be controlled due to the patterned cross-linking distribution?

Response: We appreciate the reviewer’s insightful question. In our system, the 2D-to-3D shape transformation is governed by in-plane growth-induced out-of-plane deformation—a distinct mechanism from conventional bilayer bending/twisting (which relies on active-passive material mismatches). Instead, our approach exploits localized in-plane shrinkage mismatches between halftone-patterned domains with graded crosslinking densities. According to our previous study²⁸, to form a cap-like structure using the halftone-encoded pattern G0-G9, locally required growth function $\eta = \frac{c}{(1+(b/R)^2)^2}$, where η refers to A_T/A_0 in our study, c is constant, b/R is the relative radius of each halftone pattern, b represents the variable radius. Deswelling ratios for halftone-encoded hydrogel films were 0.3 (FM-G0) ~ 0.50 (FM-G9), while swelling ratios were 1.84 (FM-G0) ~ 2.07 (FM-G9) (Fig. R9a), demonstrating a considerable variation in the cap structure that can be formed. For instance, the design principle of the accessible cap region is 65.4% (b/R) in the deswollen stage (G0-G9), and 22.8% (b/R) in the swollen stage (Fig. R9b). Therefore, even when designed with halftone gradient, the hydrogel membrane retains a relatively flat surface when fully swollen (Fig. R3a, c).

3D morphogenesis is selectively triggered by thermal stimuli, where the large deswollen mismatch between halftone-encoded regions at different grayscale levels generates non-uniform in-plane growth (Fig. R9c-f). This differential contraction converts the latent 2D pattern into programmable out-of-plane deformations. Thus, flatness in the swollen state is inherently maintained, while the deswelling process—governed by crosslinking-density-dependent phase separation—enables precise 3D shaping.

Figure R9. Design principles to encode gradients in halftone-encoded hydrogel films, facilitating the transition from 2D films to non-Euclidean 3D shapes. **a**, Swelling/deswelling ratio of hydrogel films encoded with FM and AM halftone patterns under different temperatures. **b**, Illustration of the accessible cap region using the growth function $\eta = A_T/A_0 = \frac{c}{(1+(b/R)^2)^2}$, where c is constant, b/R is the relative radius of each halftone-encoded circle, and b represents the variable radius of the dome. **c-f**, Halftone

pattern-encoded growth functions enabling the transformation of 2D hydrogel films into axisymmetric non-Euclidean 3D shapes, including spherical caps (**c**, **d**) and hyperbolic saddles (**e**, **f**).

Comment 7. The presence of cross-linking distribution would induce 3D morphologies. How does it influence the 2D display, or how can it be balanced?

Response: We thank the reviewer for raising this important point. By exploiting temperature-responsive shape transformations and optical contrast modulation, we demonstrate that encoded information can dynamically appear or disappear as 2D films transform into prescribed 3D structures (Figs. R3a–d). Our design ensures that optically encrypted information (e.g., patterns with information) is confined to specific regions of the 3D topography (~5 mm scale), while growth function-regulated gradient is encoded into the surrounding hydrogel membrane (~20 mm scale), governing 2D-to-3D shape transformation. Thus, local halftone contrast for information encoding is decoupled from the global deswelling gradient responsible for shape-morphing, enabling simultaneous control over each function.

To further mitigate potential 3D morphological perturbations induced by patterned information, we introduced a buffer region between the encrypted pattern and the surrounding hydrogel (Fig. R3d). This modification successfully eliminated potential distortions caused by localized property mismatches. Conversely, in Fig. R10, we deliberately coupled optical and shape-shifting functionalities, where the encrypted pattern actively participated in guiding 3D morphogenesis and local textures.

In our future work, we are developing a multilayer strategy to fully decouple information display and 3D shape transformation (Fig. R3l). As evidenced in Fig. R3h–k, the hydrogel's surface forms a porous layer during curing (50–120 s) to serve in the optical appearance, while the interior remains dense—a structure we hypothesize governs 2D-to-3D transformation. In this refined two-layer design, a top "skin layer" will be engineered for optical display, and the second layer will exclusively drive 3D shape morphing via encoded shrinking-induced gradients. This hierarchical architecture promises independent control over optical display and 3D structures, enabling seamless integration of multifunctional performance.

Figure R10. Shape-morphing of complex surface morphologies featuring varying local Gaussian curvatures (positive, negative, or hybrid) and halftone pattern-regulated textures.

Reviewer #2 (Remarks to the Author):

In this manuscript, Sun et al. reported a 4D printing strategy that imparts programmable control over optical appearance, mechanical properties, surface texture, and shape transformation within a single smart hydrogel film in response to various external stimuli. This approach shows potential for certain applications such as smart skins and soft robots. I recommend acceptance of this manuscript after careful revisions. Below are some comments and questions that need clarification:

Response: We sincerely thank the reviewer for the thoughtful evaluation of our manuscript and the positive recommendation for acceptance.

Q1. In the abstract, the authors mentioned that the 4D printing method enables simultaneous and programmable control over optical appearance, mechanical properties, surface texture, and shape transformation within a single smart hydrogel film. However, the manuscript lacks conclusive experimental evidence demonstrating the spatiotemporal coupling of optical, mechanical, shape-morphing, and textural responses under identical stimuli, as claimed. How to achieve programmable control over multiple properties at the same time?

Response: We appreciate the reviewer for this comment. Demonstrating simultaneous control over optical, mechanical, textural, and shape-morphing properties under identical stimuli is crucial.

This work introduces a versatile halftone-encoded 4D printing platform capable of digitally programming multifunctional properties—optical appearance, mechanical stiffness, and shape-morphing behavior—within a single synthetic hydrogel system. Through binary domain encoding—“0” for lightly crosslinked regions and “1” for highly crosslinked regions—our method enables spatially controlled co-design of disparate physical properties without the need for multiple inks or multimaterial switching. For example, these binary domains demonstrate different Young’s modulus (40 kPa vs. 123 kPa), optical appearances (opaque vs. transparent), and shrinkage ratios (0.3 vs. 0.5). By spatially arranging these domains and their integration for forming various grayscale levels, we can program the local and global multiphysical behaviors of the printed construct (Fig. R1). More importantly, this strategy enables a unique halftone-encoded 4D printing platform for co-designing multifunctionality within a monolithic hydrogel system. The following sections elaborates on our co-design strategy for multifunctional controls.

Co-design of optical and mechanical features. Our halftone-encoded approach enables simultaneous and spatially resolved control over local mechanical and optical properties. The cononsolvency-induced phase separation and pore formation in the surface skin layers enable dynamic modulation of optical contrast, while the mechanical heterogeneity introduced via halftone encoding allows for tunable strain distributions, which we visualized through full-field DIC strain mapping (Figs. R2). For instance, after immersing the printed hydrogel in ethanol followed by water at 35 °C, the optical information was erased due to solvent-induced opacity change (Fig. R2n). Yet, strain mapping retained a clear mechanical image of the Mona Lisa (Fig. R2 o-p), confirming

that our approach supports multi-modal encryption and decryption based on orthogonal physical properties.

Co-design of optical and shape-morphing features. By exploiting temperature-responsive shape transformations and optical contrast modulation, we demonstrate that encoded information can dynamically appear or disappear as 2D films transform into prescribed 3D structures (Figs. R3a-d) with controlled textures (Figs. R3f-k). Importantly, local halftone contrast for information encoding and the global shrinkage gradient responsible for shape-morphing can be simultaneously controlled for multifunctions. In future work, we plan to develop a two-step printing process to introduce an artificial skin layer that fully decouples the two functions: a thin top layer dedicated to image display and revelation, and a secondary layer responsible for shape transformation (Figs. R3l).

To better highlight the novelty and underlying scientific mechanisms of this co-design platform, we have expanded the mechanistic discussions in the revised manuscript and added more discussions summarizing these key novelties, limitations, and future perspectives.

Q2. In the introduction, the authors claimed capabilities beyond existing synthetic materials. Authors should provide a comparative evaluation to highlight the contributions of this work, helping readers understand its significance.

Response: We thank the reviewer for this comment. In response to the recommendation, we have expanded the Conclusion section to include detailed discussions emphasizing the multifunctional control, co-design strategy, the limitations, and future perspectives of our approach (see details in the response to Q1). A comparison of various functional materials and structures has also been included in the supplementary table 1 to highlight the advantages of our halftone-encoded smart hydrogel system.

Q3. In the text (line 48-49, page 3), it is written that “However, achieving refined, multi-faceted control over property and shape changes in synthetic materials remains a significant challenge...”.

I think a deeper summary of the challenges in this field is necessary, as it will help readers understand the innovations of this work.

Response: We thank the reviewer for this comment. We have added more detailed discussions in the revised manuscript, which is also included below.

“**Line 49:** However, achieving such refined, multi-faceted control in synthetic materials remains a significant challenge. Nanocomposites offer tunable optical and mechanical properties but generally lack dynamic reconfigurability⁸⁻¹⁰. Shape memory polymers (SMPs) enable programmable shape changes yet provide limited optical functionality¹¹⁻¹⁵. Liquid crystal elastomers (LCEs) combine optical anisotropy with stimuli-responsiveness but struggle to achieve true multi-modal control^{16, 17}. Smart hydrogels have emerged as promising alternatives due to their intrinsic stimulus-responsiveness and tunable optical properties¹⁸⁻²³. Nonetheless, these synthetic material systems still fall short of realizing simultaneous and coordinated control over diverse dynamic features within a single construct (see Supplementary Table 1)⁸⁻²³.”

Q4. It is high effective to use a 4D printing technique to achieve precise spatiotemporal control over photopolymerization within smart hydrogel films. However, further details on how the experimental procedure was devised are required, and the authors should provide a more comprehensive description of the methodology.

Response: We thank the reviewer for this insightful comment. In response, we have added additional information to the Methods section as well as a new Supplementary Fig. 15, clarifying both the experimental setup and the rationale behind the procedure design. A summary is provided below.

Digital halftone stereolithography was carried out using an Anycubic Photon D2 digital light processing (DLP) 3D printer (UV light intensity: 2.5 mW cm^{-2} , wavelength: 405 nm). The projection cell was assembled by placing a polydimethylsiloxane (PDMS) spacer (thickness: 450–550 μm) on a Teflon FEP film. After degassing the resin solution with nitrogen, it was injected into the cell and sealed with a glass coverslip (Brain Research Laboratories, MA). This setup employs a dynamic mask, generated by the Digital Micromirror Device (DMD) of the DLP printer (resolution: $\sim 50 \mu\text{m}/\text{pixel}$), to encode halftone patterns for precise spatiotemporal control of photopolymerization.

To achieve continuous-tone imaging, various halftoning algorithms (e.g., Fig. 3a, b, and Fig. 5a, e, i) were used to generate ten discrete grayscale levels within a 6×6 -pixel unit matrix. As an illustrative example, the Mona Lisa image (7000×7000 -pixel RGB, Fig. 3c) was first converted to a 120×120 -pixel grayscale image with 256 grayscale levels (Fig. 3d). These grayscale values were then uniformly binned into ten intervals, corresponding to the grayscale levels represented by predefined halftone patterns labeled G0 to G9. The resulting image was transformed into a 720×720 -pixel halftone pattern and processed in MATLAB to generate stereolithography (STL) files.

Notably, the STL files encode binary pixels by modulating the relative pixel heights: black pixels with lower heights correspond to shorter UV exposure durations (e.g., 50 s), while white pixels with greater heights correspond to longer exposures (e.g., 120 s) (Fig. R11). These STL files were subsequently sliced into 2D projection layers using Anycubic Photon Workshop software for DLP printing.

Figure R11. Exposure time distribution in a STL file represented by the pixel heights: “0” and “1” indicate different exposure times (such as 50-s and 120-s), the thickness of hydrogel films was restricted by using the printed cell (thickness: 450–550 μm) and glass coverslip.

Q5. The rationale for selecting UV exposure times requires further clarification. The choice of only two exposure times (50 s and 120 s) appears arbitrary. Expanding the tested range (e.g., 80 s, 100 s, and 140 s) would strengthen the structure-property correlation between exposure times and material performance.

Response: We sincerely thank the reviewer for this comment. According to review’s recommendation, the optical microscopic images, swelling/deswelling ratios, mechanical properties of hydrogel films under different curing times have been added to Supplementary Information (Supplementary Figs. 2, 8a, 12a), and more explanation has been added to the main manuscript, which are also briefly summarized below.

Our selection of 50-s and 120-s as key thresholds was guided by the need to balance structural fidelity, functional performance, and reproducibility. At 50-s exposure, the hydrogel reaches sufficient crosslinking density to maintain mechanical stability while exhibiting an opaque appearance. Further exposure up to 120-s gradually reduces this opacity, resulting in a transparent state, as confirmed by microscopic imaging (Fig. R5a). This transition window (50–120 s) was deliberately chosen to avoid undercuring (<50-s), which compromises structural integrity, or overcuring (>120-s), which leads to excessive brittleness (Fig. R5c).

Moreover, the shrinkage ratio—a critical parameter for 2D-to-3D shape transformation—scales predictably with exposure time, ranging from ~ 0.3 ($(A_{35^\circ\text{C}}/A_0)_{50\text{-s}}$ at G0) to ~ 0.5 ($(A_{35^\circ\text{C}}/A_0)_{120\text{-s}}$ at G9) (Fig. R5b). This controllable deformation range allows precise spatial tuning of the hydrogel’s in-plane growth during deswelling-induced shrinkage. Importantly, mechanical characterization (Fig. R5c) revealed that properties such as modulus and toughness evolve continuously across this exposure range, with performance peaks aligning with the 50–120 s window. Deviations outside this range disrupt the coupling between optical, mechanical, and shape-morphing behaviors, which are essential for the system’s multifunctionality. Thus, these thresholds represent optimized conditions to achieve the desired dynamic responses while ensuring reproducibility.

According to the reviewer’s recommendation, the microscopic images, stress-strain curves, and areal swelling/deswelling ratio ($A_{35^\circ\text{C}}/A_0$) of hydrogel films modulated solely by curing times have been added to the Supplementary information (Supplementary Figs. 2, 8a, 12a). More explanation has been added to the main manuscript, and is briefly summarized below.

“Line 118: We first examined variations in the optical transmittance of hydrogel films printed with varying UV exposures under different swelling-deswelling states (Supplementary Figs. 1 and 2). ...

Line 134: However, upon heating to 35 $^\circ\text{C}$ (above the LCST), the films deswelled, exhibiting distinct optical appearances in their fully deswollen states: lightly crosslinked

hydrogels (50-s) turned opaque; opacity gradually diminished with longer curing times (60-s – 110-s); and highly crosslinked hydrogels (120-s) became transparent (iv in Fig. 2a and Supplementary Fig. 2). This progression generated a striking black-and-white visual contrast against a black background (iv in Fig. 2a, Supplementary Figs. 1d and 2).”

“**Line 414:** Using the FM method to encode grayscale levels from G1 to G9, we achieved areal deswelling ratios ($A_{35^{\circ}\text{C}}/A_0$) ranging from approximately 0.3 to 0.5, as depicted by the projected blue curves (Fig. 6a, c). These halftone-encoded grayscale levels regulated deswelling-induced deformations in a manner similar to those observed under varying curing times (Supplementary Fig. 12).”

Minor points:

Q1. Figure 1: Some of the scale bars are difficult to make out from the background color. Please use different colors to improve the contrast.

Response: We thank the reviewer for this comment. We have improved the quality of all figures to make the scale bar easier to access.

Q2. Supplementary Fig 7: The unit format appears inconsistent: strain (%) is used in Fig. S7a, whereas strain is shown without units in Figs. S7b, c and Fig. 4. Please use a consistent format.

Response: We thank the reviewer for this comment. We have unified all units in these figures (e.g., the new Supplementary Fig. 8).

Reviewer #3 (Remarks to the Author):

The current paper introduces a halftone-encoded 4D printing method that enables simultaneous and programmable control over optical appearance, mechanical properties, surface texture, and shape transformation within a single smart hydrogel film in response to external stimuli, including temperature, solvents, and mechanical stress. The resolution and versatility of the proposed method are impressive. The reported data and results are solid. Publication of the paper in NC is recommended. The authors are suggested to consider the following comments to further strengthen the paper:

1. The authors may want to discuss the timescales for transitions under different conditions. Additionally, the authors may discuss potential future works when fast-transition applications are required.

Response: We sincerely thank the reviewer for this comment. We have provided some details of the timescales for the transition under different conditions, including:

“**Line 238:** However, upon immersion in ice water, the image gradually emerged within 60-100 seconds, revealing white contrasts in the brighter-tone regions.

Line 246: More importantly, the halftone-encoded hydrogel film demonstrated repeatable and reversible image concealment and revelation within 60-100 seconds through an ethanol-water immersion cycle.

Line 261: To achieve faster dynamic decryption in water, an accelerated heating protocol was implemented across the LCST, reducing the decrypted time from 7.5 hours (blue Route-1, Fig. 3g) to 3 hours (red Route-2, Fig. 3g). Initially, the image appeared blurred due to rapid phase separation and retained water within the network. Maintaining the temperature above the LCST for an additional hour enhanced image clarity as most of the water gradually diffused out. Additionally, quenching the deswollen hydrogel film in ethanol rapidly erased the image, rendering the entire film transparent (Supplementary Fig. 7).

Line 441: Following the thermal process Route-1, from room temperature to 35°C, the thermo-responsive hydrogel “skins” morphed into their prescribed non-Euclidean 3D shapes, with halftone-pattern-regulated Gaussian curvatures and surface textures (Supplementary Fig. 13). The primary shape transition occurred around LCST and continued till 35°C.”

In our future work, we will explore hierarchically structured hydrogels with highly interconnected pores for applications requiring rapid response. Such microstructures have been demonstrated to enhance solvent diffusion while enabling tunable optical and mechanical properties (added Adv. Mater. 2021, 33, 2008235 as Ref. 31). We also add it to the discussion and conclusion section in the revised manuscript.

2. What is the minimum feature size (pixel) at which halftone fidelity breaks down due to reasons such as radical diffusion or overcuring?

Response: We sincerely thank the reviewer for this insightful question. Radical diffusion and overcuring have been carefully considered in the preparation of the hydrogel films. To minimize cross-contamination between adjacent regions, lightly crosslinked areas (“0” domains) were cured for 50 seconds, while highly crosslinked areas (“1” domains) were cured for 120 seconds—conditions optimized to prevent undercuring or overcuring within each domain. To ensure pattern fidelity during photopolymerization, the minimum domain size was set to 2 × 2 pixels. This threshold was selected to avoid excessive radical diffusion that could compromise resolution at smaller scales. Under these curing conditions, the as-printed dimensions of both “0” and “1” domains matched the designed pixel size. Following gradual heating and deswelling at 35 °C, the domain sizes diverged due to differences in shrinkage (deswelling) ratios: $(A_{35^{\circ}\text{C}}/A_0)_{50-s} = 0.3$ vs. $(A_{35^{\circ}\text{C}}/A_0)_{120-s} = 0.5$. For example, in grayscale level G1, the “0” domain exhibited a post-heating (deswollen) feature size of approximately 47 μm, while the “1” domain measured ~72 μm. In higher grayscale levels such as G8, the “0” and “1” domains measured ~42 μm and ~85 μm, respectively. However, when the domain size was reduced to 1 × 1 pixel, particularly in high grayscale levels (G7–G8), the fidelity of the “0” domain was noticeably compromised due to diffusion from adjacent “1” domains. This observation reinforces the necessity of maintaining a minimum domain size of 2 × 2 pixels to preserve pattern integrity.

To quantify this effect, we characterized the optical contrast of halftone-encoded hydrogel films with varying domain sizes and performed DIC strain mapping under external load (Fig. R12). The study employed halftone patterns of identical grayscale levels but differing in feature size (1×1 , 2×2 , and 3×3 pixels). These findings reveal that reduced domain size compromises both optical uniformity and mechanical response due to radical diffusion and light scattering, resulting in deviations from predictions.

Figure R12. In-situ imaging characterization of halftone-patterned films with varying feature size. Halftone patterns with checkerboard-like distribution (50% grayscale level) featuring the size of 1×1 , 2×2 , and 3×3 pixels (a), displaying the more uniform optical (b) and mechanical response (c, d) with increased feature size.

3. Is there a tradeoff between resolution and the optical/mechanical contrast achieved?

Response: We sincerely thank the reviewer for this comment. As discussed in our response to Q2, a minimum domain size of 2×2 pixels is required to ensure pattern fidelity during photopolymerization. This threshold was established to prevent excessive radical diffusion, which could compromise spatial resolution at smaller feature sizes. As a result, different halftoning algorithms encoding the same grayscale level yield nearly identical optical and mechanical properties (Figs. R1b-c, R13). For instance, when hydrogel films encoded with either frequency-modulated (FM) or amplitude-modulated (AM) halftoning methods were gradually heated to 35°C in water—surpassing the LCST—both revealed a high-contrast Mona Lisa image (Fig. R13c, d). Owing to their different pixel arrangements, FM halftoning, which uses fine, randomly dispersed dots, excels in preserving fine detail and minimizing visual artifacts, making it ideal for high-resolution applications. In contrast, AM halftoning generates smoother tonal gradients, which are beneficial for contrast-rich images but may introduce Moiré patterns in images with intricate features.

In terms of mechanical properties, deswollen hydrogel films encoded with FM and AM patterns exhibited comparable Young's modulus values at each grayscale level (Fig. R1c). Across grayscale levels from G0 to G9, the modulus increased progressively due to the higher proportion of highly crosslinked domains.

Unlike optical images, where contrast is primarily determined by halftone resolution, the mechanical information captured via strain mapping is influenced not only by resolution

but also by anisotropic mechanical responses under deformation. Specifically, the spatial distribution of strain is governed by heterogeneity in local mechanical properties. We found that adjusting the orientation of halftone patterns could modulate mechanical anisotropy and thereby alter how features appear in strain mapping images (Figs. R2a-c, e-m). However, the strain maps generally exhibit lower resolution and contrast compared to their optical counterparts. To overcome this limitation, we developed a strategy that exploits controlled anisotropy to enhance feature resolution in strain mapping. By selectively modifying the orientation of halftone patterns—particularly in shadowed or feature-rich regions such as the eyes, nose, mouth, and hair—we introduced additional mechanical heterogeneity at constant grayscale levels. When the hydrogel film was stretched along the x-axis, these anisotropic domains produced enhanced strain contrast and sharper definition of facial features (Fig. R2o-p), resulting in more distinct interfaces between regions with different mechanical responses.

Figure R13. Halftone patterns generated using the FM method (a) and AM method (b) within a 6×6 -pixel unit matrix, displaying defined grayscale levels from G0 to G9 (top panels) and corresponding printed

hydrogel films in the deswollen state (bottom panels). Hydrogel film encoded with FM-generated halftone patterns (c) and AM-generated halftone patterns (d) briefly displays nearly identical white and black contrast when placed on a black background in the deswollen state at 35 °C.

4. How many stimulus cycles can the material undergo before losing contrast?
Response: We thank the reviewer for this insightful question. We have added a set of experiments to demonstrate the stability of the hydrogel system, taking FM-modulated *Mona Lisa* as examples. After 3-4 cycles, the hydrogel can still maintain good contrast (Figs.R14, 15).

Figure R14. a-e, A halftone-encoded hydrogel film for repeatable and reversible image concealment and

retrieval through an ethanol-water immersion cycle, enabling dynamic information decryption. The information remains visible after 3 cycles.

Figure R15. A halftone-encoded hydrogel film (e.g., FM-modulated grayscale level G8 for film region, G3-G5 for PSU letters) for repeatable and reversible image concealment and retrieval through an ethanol-water immersion cycle, enabling dynamic information decryption. The information keeps visible after 3 cycles.

5. How do ambient environments impact hydrogel function? Are there encapsulation strategies that could extend lifetimes?

Response: We sincerely thank the reviewer for this insightful question. The functionality of PNIPAm-based hydrogels is highly influenced by environmental conditions, particularly humidity and temperature. Under aqueous conditions, the halftone-patterned PNIPAm hydrogel—covalently crosslinked via UV curing—demonstrates excellent long-term stability and consistent multi-responsive behavior to thermal and solvent stimuli. However, when exposed to ambient air, even at controlled temperatures, gradual dehydration occurs. This loss of water significantly diminishes the hydrogel's dynamic performance, especially its precisely tunable optical appearance and shape-morphing capabilities.

To mitigate this limitation, we recommend implementing encapsulation strategies that address the key environmental stressors: (1) maintaining sufficient humidity or aqueous surroundings to prevent dehydration, and (2) ensuring stable thermal conditions to avoid uncontrolled phase separation near the LCST. We have validated the effectiveness of these strategies in our stability studies, which show preserved hydrogel performance over extended durations or across multiple actuation cycles (see Supplementary Fig. 9d–e, and Fig. R14, 15). For applications involving prolonged air exposure, we also suggest

exploring hygroscopic additives, moisture-retaining agents (e.g., nonpolar solvents), or thin hydrophobic barrier coatings to slow water loss while maintaining responsiveness. These approaches could significantly extend the operational lifetime of the hydrogel system in non-aqueous environments.

Additionally references:

25. Bowman, C.N.; Kloxin, C.J. Toward an enhanced understanding and implementation of photopolymerization reactions. *AIChE Journal* 2008, 54(11), 2775-2795. DOI: 10.1002/aic.11678.
26. Lehmann, M.; Krause, P.; Miruchna, V.; von Klitzing, R. Tailoring PNIPAM hydrogels for large temperature-triggered changes in mechanical properties. *Colloid and Polymer Science* 2019, 297(4), 633-640. DOI: 10.1007/s00396-019-04470-0.
31. Alsaïd, Y.; Wu, S.; Wu, D.; Du, Y.; Shi, L.; Khodambashi, R.; Rico, R.; Hua, M.; Yan, Y.; Zhao, Y.; Aukes, D. Tunable sponge-like hierarchically porous hydrogels with simultaneously enhanced diffusivity and mechanical properties. *Advanced materials* 2021, 33(20), 2008235. DOI: 10.1002/adma.202008235.
32. Henschel, C.; Schanzenbach, D.; Laschewsky, A.; Ko, C.H.; Papadakis, C.M.; Müller-Buschbaum, P. Thermoresponsive and co-nonsolvency behavior of poly (N-vinyl isobutyramide) and poly (N-isopropyl methacrylamide) as poly (N-isopropyl acrylamide) analogs in aqueous media. *Colloid and Polymer Science* 2023, 301(7), 703-720. DOI: 10.1007/s00396-023-05083-4.
35. Nojoomi, A.; Jeon, J.; Yum, K. 2D material programming for 3D shaping. *Nature Communications* 2021, 12(1), 603. DOI: 10.1038/s41467-021-20934-w.
37. Eklund, A.; Hu, S.; Fang, Y.; Savolainen, H.; Pi, H.; Zeng, H.; Priimagi, A.; Ikkala, O.; Zhang, H. Bright and switchable whiteness in macro-crosslinked hydrogels. *Advanced Optical Materials* 2024, 12(11), 2302487. DOI: 10.1002/adom.202302487.